# Neuro-mimetic Task-free Unsupervised Online Learning with Continual Self-Organizing Maps

## Abstract

An intelligent system capable of continual learning is one that can process and extract knowledge from potentially infinitely long streams of pattern vectors. The major challenge that makes crafting such a system difficult is known as *catastrophic forgetting* – an agent, such as one based on artificial neural networks (ANNs), struggles to retain previously acquired knowledge when learning from new samples. Furthermore, ensuring that knowledge is preserved for previous tasks becomes more challenging when input is not supplemented with task boundary information. Although forgetting in the context of ANNs has been studied extensively, there still exists far less work investigating it in terms of unsupervised architectures such as the venerable self-organizing map (SOM), a neural model often used in clustering and dimensionality reduction. While the internal mechanisms of SOMs could, in principle, yield sparse representations that improve memory retention, we observe that, when a fixed-size SOM processes continuous data streams, it experiences concept drift. In light of this, we propose a generalization of the SOM, the continual SOM (CSOM), which is capable of online unsupervised learning under a low memory budget. Our results, on benchmarks including MNIST, Kuzushiji-MNIST, and Fashion-MNIST, show almost a two times increase in accuracy, and CIFAR-10 demonstrates a state-of-the-art result when tested on (online) unsupervised class incremental learning setting.

## 1 Introduction

A major hurdle in designing autonomous, continuously learning agents is ensuring that such systems effectively preserve previously acquired knowledge when faced with new information. The difficulty that agents face in retaining information acquired over time is known as catastrophic forgetting (or interference) McCloskey & Cohen (1989); Ratcliff (1990); McCloskey & Cohen (1989) and is a significant challenge in the problem setting known as lifelong or continual learning Thrun & Mitchell (1995); Chen & Liu (2016); Masana et al. (2023); Ororbia et al. (2020; 2022). In continual learning, an agent must embed knowledge learned from data in a sequential manner without compromising prior knowledge. Much as humans do, this agent should process samples from these sources online, consolidating and transferring the knowledge acquired over time without forgetting previously learned tasks.

Notably, catastrophic interference has been investigated with respect to deep neural networks (DNNs), which are often trained to solve supervised prediction tasks, especially in efforts such as Kirkpatrick et al. (2017); Lopez-Paz & Ranzato (2017b); Shin et al. (2017); Ororbia et al. (2020; 2022). However, a relative dearth of work exists concerning less mainstream neural systems, particularly those that conduct unsupervised learning such as self-organizing maps (SOMs) Kohonen (1982). In this work, we seek to rectify this gap by, first, studying the degree to which forgetting occurs in a classical model such as the SOM and, second, developing a generalization of this system, which we call the Continual SOM or Continual Kohonen map (*CSOM*), which is robust to the interference encountered in the context of online continual learning.

In essence, SOMs are a type of brain-inspired (or neuro-mimetic) unsupervised neural system where its neuronal units compete for the right to activate in the presence of particular input patterns, and the synaptic parameters associated with the winning unit(s) are adjusted via a form of Hebbian learning Hebb et al.

(1949); Martinetz (1993); Ororbia (2023). Notably, the SOM's units are often arranged in either a spatial format, e.g., in a Cartesian plane/grid, or in a topological fashion, e.g., in a neighborhood/field based on the Euclidean distance between the activation values of units themselves. A useful property of the SOM is that its topologically-arranged neuronal units effectively learn to construct a low-dimensional "semantic" map of the more complicated sensory input space, where similar data patterns are grouped more closely together (around particular "prototypes") and farther apart from more dissimilar ones. This makes SOMs quite useful for clustering and dimensionality reduction tasks Vesanto & Alhoniemi (2000); Bação et al. (2005); Bigdeli et al. (2022).

At first glance, it would appear that a competitive learning model such as the SOM might offer a natural immunity to forgetting since any particular neuronal unit tends to activate more often than others (by virtue of the distance function) in response to similar sensory input patterns throughout the learning process. This specialization would, in principle, result in non-overlapping, sparse representations, which have been argued to be a key way for reducing neural cross-talk Srivastava et al. (2013); Ororbia (2021), a source of forgetting McCloskey & Cohen (1989); Ratcliff (1990); French (1999). Furthermore, some approaches, such as Gepperth et al. (2015), have advocated for using SOMs in the continual learning setting. Nevertheless, as uncovered in the experiments of this paper, the SOM, in its purest form, appears to be prone to forgetting, thus motivating our particular model generalization and computational framework Mermillod et al. (2013).

In service of the problem of online continual learning, the core contributions of this paper can be summarized as:

- We adapt the classical SOM model, which was initially formulated for fitting to single static datasets, to the online continual learning setting; specifically, we study its memory retention in tandem with its ability to adapt to pattern streams. Our experiments show that the SOM experiences substantial interference across all benchmarks examined.

- We develop a generalization of the SOM, termed the CSOM, which experiences significantly less forgetting by introducing mechanisms such as specialized decay functionality and running variance to select the best matching unit (BMU) for an input at a given time step.

- We present experimental results for four class-incremental datasets, i.e., MNIST, Fashion MNIST, Kuzushiji-MNIST, and CIFAR-10, and empirically demonstrate the robustness of the proposed CSOM to forgetting, yielding a promising, unsupervised neuromimetic system for the lifelong learning setting.

## 2   Related Work

In this section, we review prior work related to the central topics that drive this paper: general principles of competitive learning, self-organizing maps, and continual unsupervised learning.

**Competitive Learning.** In competitive learning Hartono (2012), neurons compete with each other to best match with encountered input sample patterns. This form of neural computation is often built on top of Hebbian learning Hebb et al. (1949); Choe (2013); Ororbia (2023) (where an adjustment made to a synapse depends on only local information that is available to it both spatially and temporally) and is typically used to perform clustering on input data. Vector quantization Gray (1984) and self-organizing maps (Kohonen maps) Kohonen (1982) are prominent examples within this family of models. Unlike modern-day DNNs, where all internal neurons participate in every step of inference and learning, in competitive learning, only the neurons that satisfy certain criteria "win" the right to compute Srivastava et al. (2013) and update their connection weight strengths. This form of learning can be useful in identifying and extracting useful features within a dataset. According to Rumelhart et al. (1986); Ororbia (2023), three fundamental elements define competitive learning in general: **1**) the model starts with a set of units that are highly similar, except for some random noise, which makes each of them respond slightly differently to a set of inputs; **2**) there is a limit to the "strength" of each unit (which motivates the notion of neighborhood functions, as discussed later); and **3**) the units are allowed to compete in some way for the right to respond to a particular subset of inputs.

**Self-Organizing Maps.** The self-organizing map (SOM) Kohonen (1982), and its many variants Khacef et al. (2020); Rougier & Detorakis (2021); Kopczyński (2021); Gliozzi & Plunkett (2018), is an unsupervised clustering neural model that adjusts its connection strengths via a Hebbian update rule Hebb et al. (1949). During training, spatially arranged clusters gradually form around the best-matching neurons within the SOM. This allows the SOM to be useful as a data exploration tool and even as an effective minimally supervised learner Lyu et al. (2024), wherein its internal units represent a summary of the latent patterns found within a dataset. In contrast to clustering algorithms, such as K-means Forgy (1965); Lloyd (1982), SOMs perform a soft clustering over inputs, which means that the connection weight update for the best matching neuron is the strongest while the updates made to the others decay/fade gradually as one moves farther away to other neurons within the best matching neuron's neighborhood.

SOMs, in the view of this study, make potentially invaluable memory systems of sensory input patterns given that they iteratively compress their information into compact parameter vectors (or neural "templates") as opposed to the raw data buffers Boschini et al. (2023); Wang et al. (2023) generally used for storing clusters or exemplars of specific task datasets, much like those often used in many continual learning methods Hayes et al. (2019); Lopez-Paz & Ranzato (2017b). Furthermore, as has been shown in prior work Bashivan et al. (2019); Pinitas et al. (2021b), the SOM's learning process is better equipped to facilitate the capture of the subtle differences (or variance) across a task's constituent data patterns.

**Continual Unsupervised Learning.** One of the applications of continual learning is to design agents capable of performing intelligent operations on low-resource or edge-computing devices, such as those generally found in self-driving cars Liu et al. (2019) or robotic control systems. Like many other real-world sources, self-driving cars generate enormous quantities of data that are often unlabeled or unannotated. Therefore, the value of unsupervised learning is relatively high for these kinds of applications. A consequence is that effort will be required to obtain optimized solutions to tackle the problem of catastrophic forgetting in unsupervised learning.

Crafting probabilistic generative models Shin et al. (2017); Ye & Bors (2023) that are capable of synthesizing data samples, which can be used to refresh the memory of task-specialized neural models (e.g., a classifier/regressor), is one prevalent approach. Ayub & Wagner (2021) employed (neural) encoder models to store the centroids of task data, which were later used to generate data from previous tasks in order to induce replay. However, this approach is not computationally feasible for large quantities of data – storing multiple encoder models per task results in an increase in memory complexity as well as incurs additional computational time needed to train each additional new encoder network for every newly encountered task. This greatly hinders the scalability of such an approach. In general, rehearsal-based approaches Rebuffi et al. (2017); Isele & Cosgun (2018), of which some schemes could technically be considered to fall under the umbrella of unsupervised continual learning, often store data samples in some form of explicit memory. However, low-resource devices often do not have a large memory capacity to store (enough) samples that adequately capture the variance of every task's distribution; therefore, this reduces such schemes' effectiveness given that explicitly storing more data to facilitate effective (memory-based) refreshing/retraining of more complex neural predictor model is not feasible Li et al. (2023).

In this work, we will show that our proposed CSOM can readily and usefully capture the variance inherent to each task's dataset (within a sequence) and yet not need constant, expensive retraining or refreshing itself. Finally, although out of scope for this paper, we remark that the SOM models we study could also be made to expand much as in (growing, Xu et al. (2022); Ye & Bors (2023); Yang et al. (2023)) dynamical neural gas models Fritzke (1994); Ventocilla et al. (2021)

## 3 Methodology

**Notation.** We start by defining the notation that will be used throughout this paper. $\odot$ indicates a Hadamard product, $\cdot$ indicates a matrix/vector multiplication (or dot product if the two objects it is applied to are vectors of the same shape). $||\mathbf{v}||_p$ is used to represent the $p$-norm (distance function w.r.t. the difference between two vectors), i.e., $p = 2$ selects the 2-norm or Euclidean distance. $\mathbf{W}[:, i]$ is the slice operator, meant to extract the $i$th column vector of matrix $\mathbf{W}$; $\mathbf{W}[i, :]$ is meant to extract the $i$th row. $\cos(\theta)$ indicates cosine similarity.

---

**Algorithm 1** [Rebuttal edit: The inference and learning processes for the (classical) SOM, formulated for the online stream setting.]

---

**Input**: sample $\mathbf{x}_i^{(k)}$, synaptic weights $\mathbf{M}$, topology $\mathcal{G}$, simulation time step $t$
**Parameter**: $\lambda, \sigma, \tau_\lambda, \tau_\sigma, \lambda_{t=0} \leftarrow \lambda, \sigma_{t=0} \leftarrow \sigma$

> **function** UPDATE($\mathbf{x}_i^{(k)}$, $\mathbf{M}$, $\mathcal{G}$, $\sigma_t$, $\lambda_t$)      ▷ Compute L2 distances and perform weighted updates
>> $\mathbf{\Delta_2} = \mathbf{x}_i^{(k)}(t) - \mathbf{M}$, $\mathbf{d} = ||\mathbf{\Delta_2}[:,j]||_2, j = 0, 1, ...H$    ▷ Compute the BMU, $u$ and its neighbors, $\mathcal{N}_u$
>> $u = \arg\min_{j \in H} \mathbf{d}$, $\mathcal{N}_u = $ GETNEIGH($u, \mathbf{M}, \mathcal{G}$)
>> **for** $v_j \in (u \cup \mathcal{N}_u)$ **do**      ▷ Update synapses of the BMU and its neighbors
>>> $\mathbf{M}[:,v_j] \leftarrow \mathbf{M}[:,v_j] + \lambda_t \phi(u, v_j, \mathcal{G}, \sigma_t)(\mathbf{\Delta}[:,v_j])$
>
> **function** TRAIN($\mathbf{x}_i^{(k)}, \mathbf{M}, \mathcal{G}, t$)      ▷ Online training routine TRAIN(.), at time step $t$
>> $\lambda_t = \lambda_0 \exp\left(-t/\tau_\lambda\right)$      ▷ or use Equation 1
>> $\sigma_t = \sigma_0 \exp\left(-t/\tau_\sigma\right)$      ▷ or use Equation 2
>> UPDATE($\mathbf{x}_i^{(k)}(t), \mathbf{M}, \mathcal{G}, \sigma_t, \lambda_t$)      ▷ inference step
>> $t \leftarrow t + 1$      ▷ Advance simulation time forward

---

**Problem Definition.** Consider a sequence of $T$ tasks, which we formally denote by $\mathcal{S} = \bigcup_{k=1}^T \mathcal{T}_k$. Each task $\mathcal{T}_k$ has a training dataset (each containing $C$ classes), $\mathcal{D}_{train}^{(k)} = \bigcup_{i=1}^{N_k}\{(\mathbf{x}_i^{(k)}, \mathbf{y}_i^{(k)})\}$, where $\mathbf{x}_i^{(k)} \in \mathcal{R}^{D \times 1}$ is a data pattern and $\mathbf{y}_i^{(k)} \in \{0,1\}^{C \times 1}$ is its label vector. Furthermore, we let $N_k$ be the number of patterns in task $T_k$ and $\mathcal{D}_{test}^{(k)}$ to represent the test dataset for task $\mathcal{T}_k$. We remark that, although the datasets investigated in this work come with labels, our models will never use them since they are unsupervised. However, we will use the labels for external analysis, as we will see in our defined metric(s). Finally, all models process each data point in $\mathcal{D}_{train}^{(k)}$ **only once (online adaptation)** and time steps are tracked in simulation within the variable $t$, i.e., each time that a data point $\mathbf{x}_i^{(k)}$ is sampled from $\mathcal{D}_{train}^{(k)}$ of task $\mathcal{T}_k$, $t$ is incremented as $t \leftarrow t + 1$.

When a continual learning agent is finished training on task $\mathcal{T}_k$ (using $\mathcal{D}_{train}^{(k)}$), the data $\mathcal{D}_{train}^{(k)}$ will be lost as soon as the model proceeds to task $\mathcal{T}_{k+1}$ (up to $\mathcal{T}_T$). Furthermore, note that $\mathcal{D}_{test}^{(k)}$ is only used for external model evaluation. The general objective/goal is to maximize the agent's generalization performance on task $\mathcal{T}_k$ while minimizing how much its performance degrades on prior tasks $\mathcal{T}_1$ to $\mathcal{T}_{k-1}$.

### 3.1 The Kohonen Map

In Algorithm 1, we present our formulation of the classical SOM (Kohonen Map) for the online processing of data points from a (task) stream. Each data point $\mathbf{x}_i^{(k)}$ is sampled from the task $\mathcal{T}_k$ (with dataset $\mathcal{D}_{train}^{(k)}$). The SOM, with $H$ neuronal units, adapts its synaptic matrix $\mathbf{M} \in \mathcal{R}^{D \times H}$ given input $\mathbf{x}_i^{(k)}$. In this study, the topology $\mathcal{G}$ is a square grid of dimensions $K \times L$ (i.e., $H = K * L, K = L$). $\phi(u, v_j, \mathcal{G}, \sigma_t)$ indicates a neighborhood weighting function used to scale the update to unit $v_j$ with synaptic vector $\mathbf{M}[:,v_j]$. Note that the weighting function depends on the euclidean distance between the BMU ($u$) and the neighboring units ($v_j$) in the cartesian plane.

Specifically, we see that inference is conducted in the UPDATE() routine of Algorithm 1. The SOM first computes the (Euclidean) distance between the current sample $\mathbf{x}_i^{(t)}$ and all internal units in $\mathbf{M}$ and then stores these distances in the distance (row) vector $\mathbf{d} \in \mathcal{R}^{1 \times H}$. The BMU is calculated by taking the argmin over $\mathbf{d}$, returning the index of the BMU as integer $u$[1] Finally, the indices/coordinates of the neighboring units of BMU – any such neighbor is indexed by $v_j$ – are returned by the GETNEIGH() sub-routine and stored in the array $\mathcal{N}_u$.

As shown in figure 1, to update the relevant synaptic weight vectors of the SOM's matrix $\mathbf{M}$, a loop runs through unit indices ($u \bigcup \mathcal{N}_u$) and a weighted Hebbian update is applied. Note that the Hebbian rule in

---

[1]Note that $u$ further maps to a fixed two-dimensional coordinate $(k_u, l_u)$, where $0 < k_u \leq K$ and $0 < l_u \leq L$, since $\mathcal{G}$ is rectangular in our formulation of SOMs.

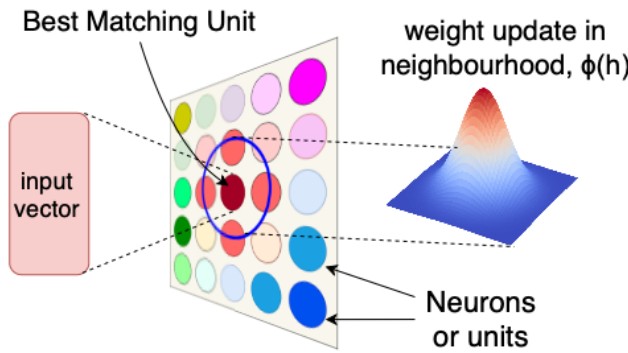

Figure 1: [Rebuttal edit: The weight update is maximum for the unit that best matches with the input. It further decays as the euclidean distance of units (in cartesian plane) increases from the best matching unit increases]

---

**Algorithm 2** [Rebuttal edit: CSOM inference and learning processes.]

---

**Require:** data sample $\mathbf{x}_i^{(k)}$, SOM weight matrix $\mathbf{M}$, running variance matrix $\mathbf{M}^{\omega^2} \in \mathcal{R}^{D \times H}$, radius $\sigma_v, \sigma_V$, learning rate $\lambda_h$, BMU count for a unit ($\eta_u$ is count for unit $u$)

1: **function** UPDATE($x_i^{(k)}, u, V, \sigma_u, \lambda_u, \mathbf{M}, \mathbf{M}^\omega$)
2:      $\delta = (2 * \sigma_u^2)^{-1}$
3:      $\tau_1 = -\delta^{-1} * \log\left(\frac{10^{-8}}{\lambda_u}\right)$
4:      **if** $\|v - u\| < \sigma_u$ **then**
5:          $\phi = \lambda * \exp(\|v - u\|^2 * \delta)$
6:          $\mathbf{M}_h = \mathbf{M}_h + \phi * (x_i^{(k)} - \mathbf{M}_h)$                 ▷ where, $h \in \{v\} \cup u$
7:          $\alpha_\sigma = min(1, \alpha_\sigma * decay\_factor)$
8:          $\sigma_h^2 = \alpha_\sigma * \sigma_h^2 + (1 - \alpha_\sigma) * \left\| x_i^{(k)} - \mathbf{M} \right\|$
9:          [Rebuttal edit: $\sigma_u = \sigma_u * \exp(-\eta_u/\tau_\sigma)$]        ▷ $\sigma_u^0$ - initial radius of $u$, $\tau_\sigma$ - decay factor
10:     [Rebuttal edit: $\lambda_u = \lambda_u * \exp(-\eta_u/\tau_\lambda)$]      ▷ $\sigma_u^0$ - initial learning rate of $u$, $\tau_\lambda$ - decay factor
11: **function** TRAIN($\mathbf{x}_i^{(k)}(t), \mathbf{M}, \mathbf{M}^{\omega^2}, \mathcal{G}, t$)
12:      $\Delta = \left\| \mathbf{x}_i^{(k)} - M \right\|_2 / \mathbf{M}^\omega$
13:      $u = \arg\min_{h \in H} \Delta$
14:      UPDATE($x_i^{(k)}, u, V, \sigma_u, \lambda_u, \mathbf{M}, \mathbf{M}^\omega$)

---

Algorithm 1 reuses the difference vectors stored in $\boldsymbol{\Delta}$ and applies both a dynamic learning rate $\lambda_t$ and a coefficient produced by $\phi(u, v_j, \mathcal{G}, \sigma_t)$. The neighborhood function $\phi(u, v_j, \mathcal{G}, \sigma_t)$ was set to be a Gaussian function, centered around the BMU indexed by $u$, with radius $\sigma_t$. Note that the **classical SOM maintains one global copy of learning rate $\lambda_t$ and radius $\sigma_t$ that is shared among all neurons**. They are designed to be dynamic, time-dependent functions and are shown in the TRAIN() routine (which also depicts the full training step performed by our online SOM). We found, through empirical study, that the following set of equations yielded better clusters as opposed to original decay functions of Kohonen (1982):

$$\lambda_t = \lambda_0(1 + t * \exp(t/\tau_\lambda))^{-1} \tag{1}$$

$$\sigma_t = \sigma_0(1 + t * \exp(t/\tau_\sigma))^{-1}. \tag{2}$$

## 3.2 The Continual Kohonen Map

In Algorithm 2, we present our proposed model, the continual SOM (CSOM [2], Figure 2), built to process and generalize dynamically from samples drawn from a stream of tasks. Notice that, first of all, the CSOM now

---

[2]All the notations are summarized in Table 4

maintains an additional (non-negative) matrix $\mathbf{M}^{\omega^2} \in \mathcal{R}_+^{D \times H}$, which contains synaptic weight parameters associated with the "running variance" ($\omega^2$) of each neuronal unit in the system. This means that each unit/prototype in the CSOM is defined by two vectors of weights – one for approximate unit means and another for approximate standard deviations. The intuition is that each unit in our SOM model is generalized to maintain its own learnable multivariate Gaussian distribution (inspired by the latent variables of incremental Gaussian mixture models) with a diagonal covariance matrix. Notice that, in the UPDATE() routine of our CSOM, the variance parameters are adjusted using a Hebbian-like rule inspired by Welford's online algorithm Welford (1962) but modified to use the weighting provided by our CSOM's neighborhood function. We denote the initial scaling/update factor for running variance of every unit as $\lambda_\omega^0$ as shown in step 11 of Algorithm 2. In SOMs, the ratio of weight update that each connection goes through depends on its cartesian distance from the BMU in a topology. In CSOM, we empirically found that the update/scaling factor of running variance of every neuron is proportional to the magnitude of its weight update. Therefore, this update factor for a neuron is decayed as per its distance from the BMU to obtain $\lambda_\omega$ in step 11 of Algorithm 2. The running variance vector coupled to every neuron in the CSOM can be used to generate samples belonging to the class that matched it. Thus, the underlying structure of our neural system could be likened to a simple, dynamic generative model.

In addition to the local running variance parameters, the CSOM is designed to promote a form of neuronal competition driven by unit-centric learning and distance weighting parameters. **Specifically, each neuron $h \in H$ arranged in topology $\mathcal{G}$ is assigned an independently-controlled, dynamic radius $\sigma_h$ and learning rate $\lambda_h$ parameter (as shown in Figure 3)**. As shown in Algorithm 2, in the routine UPDATE(), we particularly decay learning parameters only for the BMU ($u$), i.e., only $\sigma_u$ and $\lambda_u$ are decayed at time $t$. This localized decay is furthermore a function of the number of times that unit $u$ has been selected as the BMU, i.e., unit $u$ adjusts $\sigma_u$ and $\lambda_u$ as a function of its BMU count $\eta_u$. We do not decay $\sigma_u$ and $\lambda_u$

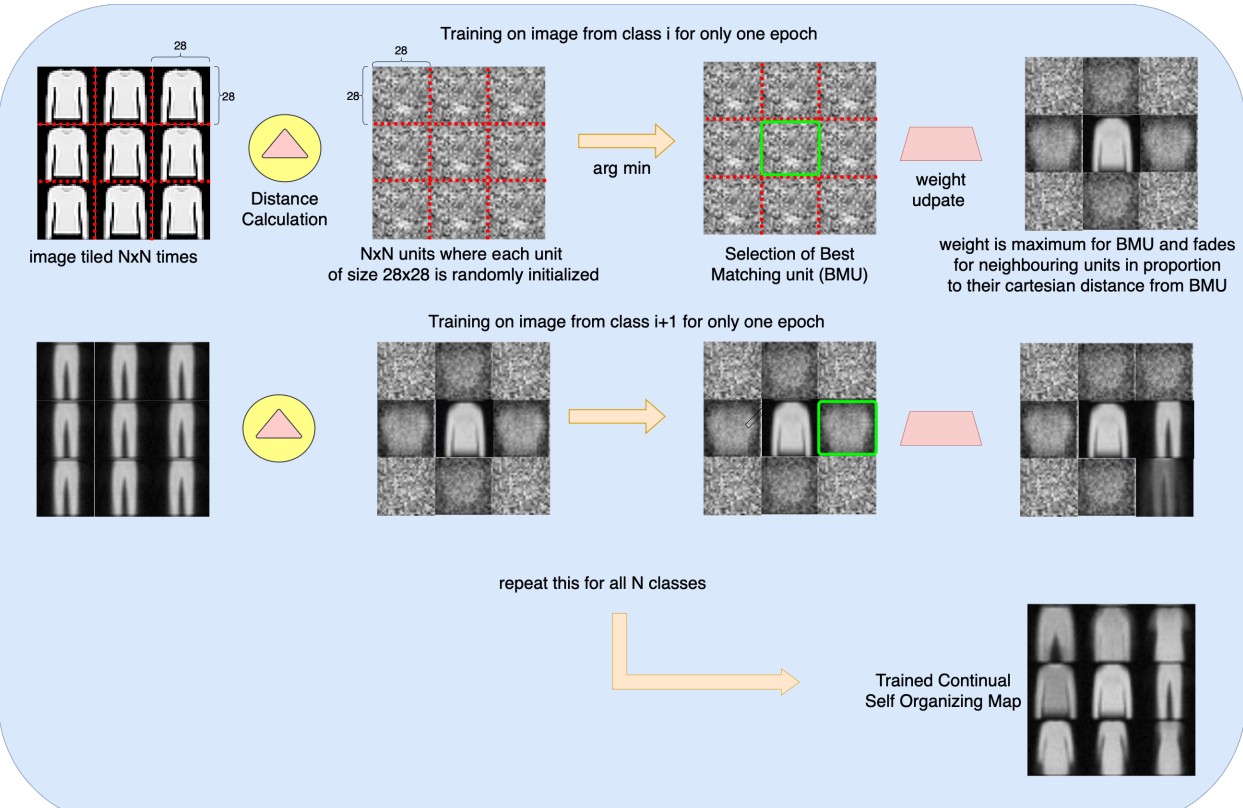

Figure 2: [Rebuttal edit: Illustration of the overall process for best-matching unit (BMU) selection and synaptic updating of the CSOM for unsupervised online learning.]

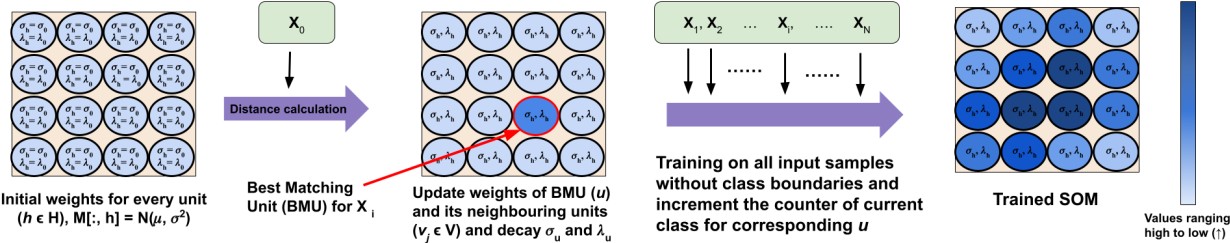

Figure 3: [Rebuttal edit: Unit centric parameter update in CSOM inspired from competitive learning]

after they reach certain infinitesimally small threshold to ensure their values do not plummet to 0 and some positive weight update is always achieved as $\eta_u \to T$. The neighborhood function of the CSOM is also notably a function of the running variance parameters $\mathbf{M}^{\omega^2}$, further facilitating the calculation of a per-unit region of influence (treating each neuron as its own weighted multivariate Gaussian distribution). As a result, whenever a weight update is triggered, synaptic values are adjusted on a linear scale for the BMU $u$ while, for any non-BMU units $v_j$, the update is adjusted on a scale that decreases as the euclidean distance from $u$ increases. The CSOM utilizes a masking matrix ($\mathbf{o}$) to control the scale of its synaptic weight updates. The mask matrix also prevents any leaky weight updates caused by infinitesimally small values passed by the neighborhood function ($\phi$). [Rebuttal edit: The L2 distance along with normalization by the running standard deviation for every input sample has time complexity of $\mathrm{O}(D \times H)$. Computing the arg min has $\mathrm{O}(H)$ time complexity whereas, weight update is again $\mathrm{O}(D \times H)$. Therefore, for $N$ input samples, the overall time complexity of CSOM will be $\mathrm{O}(N \times D \times H)$. At runtime, we maintain the CSOM weights and their respective running variance values (both, $\mathrm{O}(D \times H)$) along with the input sample of size ($D$), therefore the total space complexity remains $\mathrm{O}(D + 2(D \times H))$ which is, $\mathrm{O}(D \times H)$]

Crucially, the proposed CSOM is a task-free model Masana et al. (2023); Aljundi et al. (2018), which means that it does not require any information about task boundaries (in the form of task descriptors). In addition, the CSOM, much like our reformulation of the SOM described earlier, is constructed to process data in an online, iterative fashion, adapting its per-unit parameters as a function of simulation time.

## 4 Mathematical Analysis

In this section, we formally show that the CSOM reaches an equilibrium state or fixed point Haag et al. (1974); Stogin et al. (2024); Burton (2003), meaning that it exhibits stability, a quality that improves its generalization ability as well as helps it avoid catastrophic forgetting.

**[Rebuttal edit: System Definition and Update Rules]**

[Rebuttal edit:

**Definition 4.1 (CSOM System)** *Let $(\Omega, \mathcal{F}, \mathbb{P})$ be a probability space and $d \in \mathbb{N}$. Consider sequences:*

$$\{\mathbf{x}(t)\}_{t=1}^{\infty} \subset \mathbb{R}^d \quad \text{(input vectors)}$$
$$\{\mathbf{M}_i(t)\}_{t=1}^{\infty} \subset \mathbb{R}^d \quad \text{(synaptic weights for neuron } i\text{)}$$
$$\{\omega_i^2(t)\}_{t=1}^{\infty} \subset \mathbb{R}_+ \quad \text{(variance estimates for neuron } i\text{)}$$

*with parameters $\lambda_\omega \in (0, 1)$, learning rates $\lambda_M(t) > 0$, and a neighborhood function $\phi(t, u, i)$. The updates are:*

$$\mathbf{M}_i(t+1) = \mathbf{M}_i(t) + \lambda_M(t)\, \phi(t, u(t), i)\big(\mathbf{x}(t) - \mathbf{M}_i(t)\big)$$
$$\omega_i^2(t+1) = \lambda_\omega \omega_i^2(t) + (1 - \lambda_\omega)\|\mathbf{x}(t) - \mathbf{M}_i(t)\|^2$$

*where $u(t) = \arg\min_j \|\mathbf{x}(t) - \mathbf{M}_j(t)\|$.*

]

**[Rebuttal edit: Assumptions]**

[Rebuttal edit:

**Assumption 1 (Parameter and Input Conditions)**   *1. $\lambda_\omega \in (0,1)$ is constant.*

  *2. $\lambda_M(t) > 0$ with $\lim_{t\to\infty} \lambda_M(t) = 0$.*

  *3. The neighborhood function $\phi(t,u,i)$ satisfies $0 \le \phi(t,u,i) \le 1$ for all $t$.*

  *4. $\sigma(t) > 0$ with $\lim_{t\to\infty} \sigma(t) = 0$ (if used in $\phi$).*

  *5. Input vectors $\mathbf{x}(t) \in \mathcal{X}$ where $\mathcal{X} \subset \mathbb{R}^d$ is compact.*

  *6. $\mathbf{M}_i(1) \in \mathcal{X}$ for all $i$ (initialization).*

]

**[Rebuttal edit: Variance Sequence Convergence]**

[Rebuttal edit:

**Theorem 4.2 (Variance Convergence)** *Suppose that for each $i$:*

  *1. $\sup_{t\ge 1} \|\mathbf{x}(t)\| \le C$ almost surely,*

  *2. $\lim_{t\to\infty} \mathbf{M}_i(t) = \mathbf{M}_i^*$ almost surely,*

  *3. $\omega_i^2(1) < \infty$ almost surely.*

*Then the sequence $\{\omega_i^2(t)\}_{t=1}^\infty$ converges almost surely to a finite limit*

$$\omega_i^{2*} = (1 - \lambda_\omega) \sum_{k=1}^\infty \lambda_\omega^{k-1} \|\mathbf{x}(k) - \mathbf{M}_i(k)\|^2.$$

]

**Proof:**   [Rebuttal edit: **Step 1: Boundedness of $\|\mathbf{x}(t) - \mathbf{M}_i(t)\|^2$**]

[Rebuttal edit:  By assumption, $\|\mathbf{x}(t)\| \le C$ and $\mathbf{M}_i(t) \to \mathbf{M}_i^*$, so $\mathbf{M}_i(t)$ is eventually bounded by $K$. Hence, $\|\mathbf{x}(t) - \mathbf{M}_i(t)\|^2 \le (C + K)^2$ for large $t$, and the maximum over all $t$ is finite.]

[Rebuttal edit: **Step 2: Explicit Formula**]

[Rebuttal edit: Define $\varepsilon_t = \|\mathbf{x}(t) - \mathbf{M}_i(t)\|^2$. The recursion is

$$\omega_i^2(t+1) = \lambda_\omega \omega_i^2(t) + (1 - \lambda_\omega)\varepsilon_t$$

The solution (by induction) is

$$\omega_i^2(t+1) = \lambda_\omega^t \omega_i^2(1) + (1 - \lambda_\omega) \sum_{k=1}^t \lambda_\omega^{t-k} \varepsilon_k$$

]

[Rebuttal edit: **Step 3: Convergence of the Weighted Series**]

[Rebuttal edit: Since $|\varepsilon_k| \leq B$,

$$\sum_{k=1}^{\infty} \lambda_\omega^{k-1} \varepsilon_k \leq B \sum_{k=1}^{\infty} \lambda_\omega^{k-1} = \frac{B}{1 - \lambda_\omega} < \infty$$

Thus, the weighted average converges. ]

[Rebuttal edit: **Step 4: Pointwise Convergence**]

[Rebuttal edit: As $t \to \infty$, $\lambda_\omega^t \omega_i^2(1) \to 0$ and

$$\sum_{k=1}^{t} \lambda_\omega^{t-k} \varepsilon_k \to \sum_{k=1}^{\infty} \lambda_\omega^{k-1} \varepsilon_k$$

] [Rebuttal edit: Thus, $\omega_i^2(t) \to \omega_i^{2*}$.]    $\square$

[Rebuttal edit:

**Weight Vector Update Convergence**

]

[Rebuttal edit:

**Lemma 4.3 (Bounded Trajectories)** *Under Assumption 1, for all $t \geq 1$ and $i$, $\mathbf{M}_i(t) \in \mathcal{X}$ almost surely.*

]

**Proof:**   [Rebuttal edit:   Each update is a convex combination:

$$\mathbf{M}_i(t+1) = (1-\alpha)\mathbf{M}_i(t) + \alpha\mathbf{x}(t)$$

where $\alpha = \lambda_M(t)\phi(t, u(t), i) \in [0, 1]$, so by convexity of $\mathcal{X}$, the property is preserved by induction. ]    $\square$

**Proposition 4.4 ([Rebuttal edit: Vanishing Increments for Weights])** *[Rebuttal edit:   Under Assumption 1,*
$$\|\mathbf{M}_i(t+1) - \mathbf{M}_i(t)\| \to 0 \text{ as } t \to \infty$$
*]*

**Proof:**   [Rebuttal edit:   The increment is

$$\|\mathbf{M}_i(t+1) - \mathbf{M}_i(t)\| = \lambda_M(t)\phi(t, u(t), i)\|\mathbf{x}(t) - \mathbf{M}_i(t)\|$$

By Lemma 4.3, $\|\mathbf{x}(t) - \mathbf{M}_i(t)\| \leq \text{diam}(\mathcal{X}) < \infty$. Since $\lambda_M(t) \to 0$, the product vanishes. ]    $\square$

**Proposition 4.5 ([Rebuttal edit: Vanishing Increments for Variances])** *[Rebuttal edit: Under Assumption 1,*
$$|\omega_i^2(t+1) - \omega_i^2(t)| \to 0 \text{ as } t \to \infty$$
*]*

**Proof:**   [Rebuttal edit: From the recurrence,

$$|\omega_i^2(t+1) - \omega_i^2(t)| = (1-\lambda_\omega)|\|\mathbf{x}(t) - \mathbf{M}_i(t)\|^2 - \omega_i^2(t)|$$

By Theorem 4.2, $\omega_i^2(t)$ converges to the average of the $\|\mathbf{x}(t) - \mathbf{M}_i(t)\|^2$, so their difference must vanish in the limit.]    $\square$

**[Rebuttal edit: Strong (Almost Sure) Convergence of Weights]**

**Assumption 2 ([Rebuttal edit: Strong Learning Rate Conditions])** *[Rebuttal edit: Additionally assume:*

1. *$\sum_{t=1}^{\infty} \lambda_M(t) = \infty$, $\sum_{t=1}^{\infty} \lambda_M(t)^2 < \infty$*

2. *$\sum_{t=1}^{\infty} \lambda_M(t) \sup_{i,j} \phi(t, i, j) < \infty$*

*]*

**Theorem 4.6 ([Rebuttal edit: Strong Convergence])** *[Rebuttal edit: Under Assumptions 1 and 2,*

$$\mathbf{M}_i(t) \to \mathbf{M}_i^* \text{ almost surely for all } i$$

*]*

**Proof:** [Rebuttal edit: Sum the increments:

$$\sum_{t=1}^{\infty} \|\mathbf{M}_i(t+1) - \mathbf{M}_i(t)\| \leq \text{diam}(\mathcal{X}) \sum_{t=1}^{\infty} \lambda_M(t) \sup_{i,j} \phi(t, i, j) < \infty$$

Thus, the sequence is Cauchy and converges almost surely.] □

**[Rebuttal edit: Practical Parameter Choices]**

**Corollary 4.7 ([Rebuttal edit: Parameter Selection])** *[Rebuttal edit: The following choices satisfy the strong convergence conditions:*

1. *$\lambda_M(t) = \frac{a}{t+b}$ with $0.5 < a < 2$ and $b > 0$*

2. *$\sigma(t) = \sigma_0 t^{-c}$ with $c > 0$ and $ac < 1$*

3. *$\lambda_\omega \in (0.9, 0.99)$*

*]*

**Proof:** [Rebuttal edit: These choices ensure:

- $\sum \lambda_M(t) = \infty$ when $a > 0.5$,

- $\sum \lambda_M(t)^2 < \infty$ when $a < 2$,

- $\sup_{i,j} \phi(t, i, j) \leq 1$, so $\sum \lambda_M(t) \sup_{i,j} \phi(t, i, j) < \infty$.

] □

[Rebuttal edit: Our analysis shows that CSOM achieves equilibrium through vanishing increments in both weight and variance components, with all steps and intermediate arguments preserved for clarity.]

## 5 Experiments

To test our proposed CSOM and determine its performance on competitive datasets, we performed experiments across three databases under two continual learning setups.

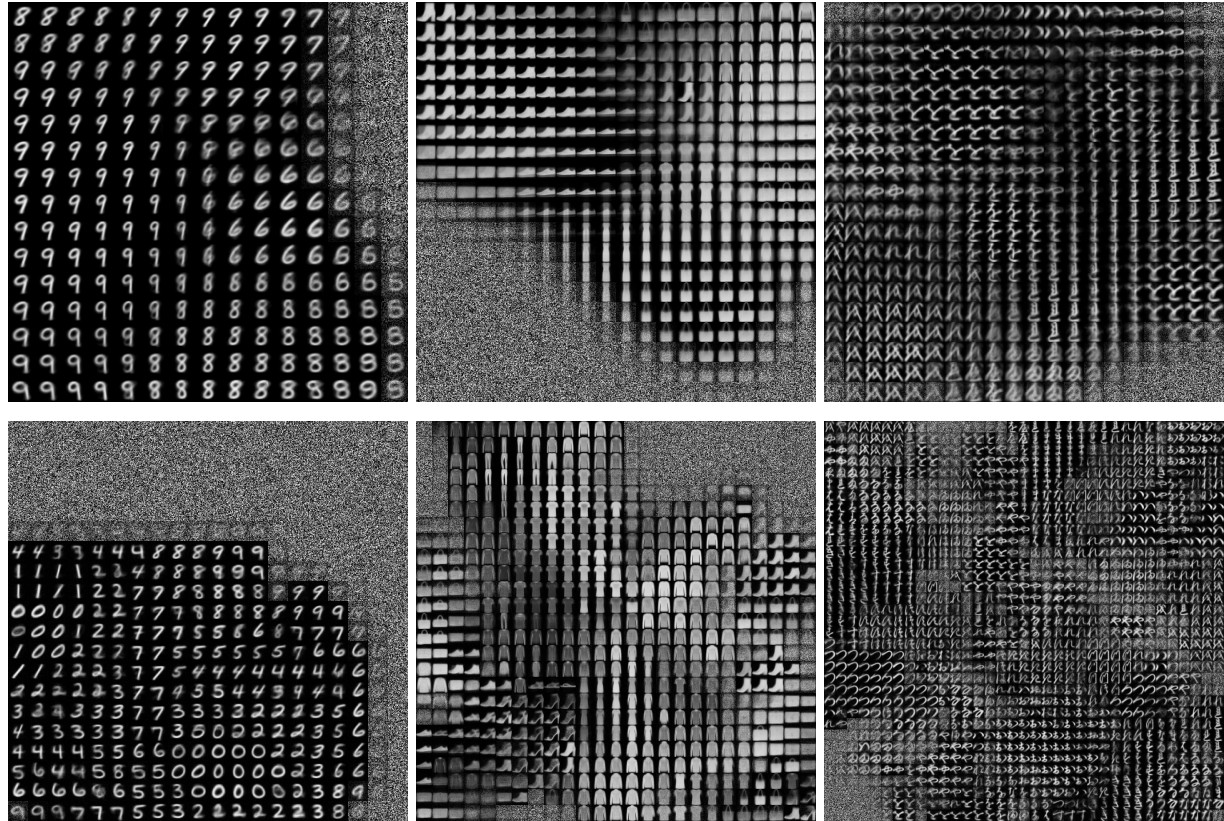

Figure 4: Class-incrementally adapted classical SOM (top row) versus continual SOM (bottom row) on: MNIST (Left), FashionMNIST (Middle), and KMNIST (Right).

## 5.1 Datasets

To evaluate the CSOM, we employed three grayscale datasets along with one containing natural image samples.

**Split MNIST, Fashion MNIST, and KMNIST:** We used variations of MNIST to test our neural models – specifically, the original MNIST database LeCun et al. (1998), Fashion MNIST (FMNIST) Xiao et al. (2017), and Kuzushiji-MNIST (KMNIST) Clanuwat et al. (2018), all containing $28 \times 28$ gray-scale pixel images. Furthermore, we transformed these datasets by normalizing them, i.e., diving them by 255.0, which brought them into the value range of $[0.0, 1.0]$, thus making the learning process easier for the SOMs/CSOM models.

**Split CIFAR-10:** To test the CSOM on images belonging to different (more complex) distributions, we also utilized the CIFAR-10 dataset Krizhevsky & Hinton (2010), which contains a large number of samples with $32 \times 32$ Red-Green-Blue (RGB) images. We specifically converted the CIFAR-10 images to grayscale to test the neural models of this study (and defer the use of separate color channels for future work). Furthermore, we normalized the images as in the MNIST datasets, bringing their pixel values into the range of $[0.0, 1.0]$.

**Model Baselines:** We adapt the Dendritic SOM (DendSOM) Pinitas et al. (2021a), further utilizing their label prediction method to evaluate the performance of our proposed model. The DendSOM uses a hit matrix to determine the label of a trained unit in the evaluation phase. We maintain a similar hit matrix (BMU count, $\eta_u$) later used in the evaluation phase for label prediction. This ultimately helps us to establish a fair comparison between our model and baselines.

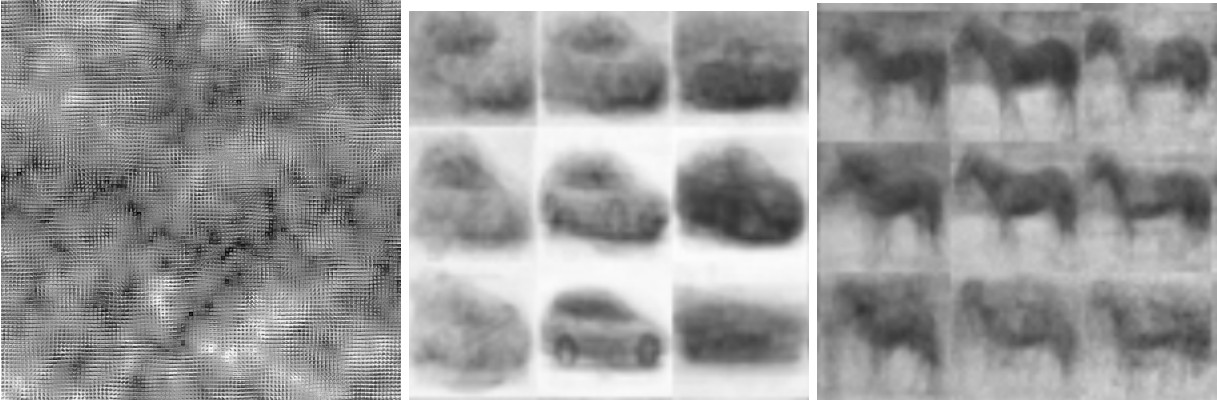

Figure 5: (Left) CSOM containing 100x100 neurons/units trained class incrementally on grayscale images of CIFAR-10. (Right) Two snapshots of trained CSOM clusters

## 5.2 Learning Simulation Setups

**Class Incremental Learning:** In this study, we focus on the continual learning setup where tasks consist of data points belonging entirely to one single class. Specifically, this means that for any given task $\mathcal{T}_t$, its training dataset $\mathcal{D}_{train}^{(t)}$ contains all data points labeled under one specific class $c \in C$ (where $C$ is the total number of unique classes in the entire benchmark dataset). For instance, for the case of Split-MNIST, this means that task $\mathcal{T}_0$ consists entirely of patterns with the class label of the digit zero, and task $\mathcal{T}_1$ consists entirely of the digit one, and so on and so forth. Furthermore, note that all models and baselines studied are trained under the same setup with the same seeding/shuffling of data points in order to ensure a fair comparison. All models are trained to process the data points incrementally/online, one sample at a time, and no data points are ever revisited (meaning that each model is only allowed one single epoch or pass through an entire task data subset, effectively simulating the online streaming learning setting Agarwal et al. (2008); Beyazit et al. (2019)).

**Domain Incremental Learning:** In this setup we simulated a domain incremental learning within a given dataset. This is similar to a task incremental setup; however, here, we divided the datasets into five tasks, where each task had data samples representing two digits, labeled in binary, i.e., $c \in [0, 1]$ (0 encoded class one, 1 encoded class two).

## 5.3 Training and Architecture Parameters

We followed a square design for the topology underlying all of the models examined in this work. This means that every SOM/CSOM had $K \times L$ units in its topology, $(K = L)$. In Table 5 (Appendix), the column for $(K = L)$ indicates three different trials with values in each trial for $K$ in a model trained on MNIST, Fashion-MNIST, KMNIST, respectively. The initial value of the running variance for all pixels in every SOM unit $(\omega_0^2)$ was decided based on the distribution of pixels in the input dataset. We found empirically that setting $\omega_0^2$ slightly higher than the variance of input distribution gave us a good starting point. Similarly, setting $\lambda_\omega^0$ within the range $[0.9, 0.99]$ helped to maintain an effective balance between the old and new values of the running variance at every simulation step. For the DendSOM, we modeled/instantiated four copies of SOM $(M)$ where the unit topology size in each copy was $[14 \times 14]$. Since the DendSOM breaks the input image into patches for every copy of SOM, the easiest way to divide an MNIST-type of image having dimensions $[28 \times 28]$ would be to have its four patches be of shape $[14 \times 14]$. We followed this for the implementations of DendSOM presented in our work. Through a brute force approach, we found that setting $\tau_\sigma = 8$ and $\tau_\lambda = 45$ performed a gradual decay of $\sigma$ and $\lambda$ throughout training.

### 5.4 Evaluation

For model evaluation, we adapted the point-wise mutual information (PMI) Pinitas et al. (2021a) measure, which loosely follows the Hebbian rule of learning to suit the design logic of CSOM. Formally, the PMI is calculated as follows:

$$PMI(l; BMU) = \log \frac{P(l|BMU)}{P(l)} \tag{3}$$

$$P(l|BMU) = \frac{\eta[l, BMU]}{\Sigma_{i \in Labels} \eta[i, BMU]} \tag{4}$$

$$P(l) = \frac{\Sigma_{h \in Units} \eta[l, h]}{\Sigma_{h \in Units} \Sigma_{i \in Labels} \eta[i, h]} \tag{5}$$

$$Predicted\ Label = \underset{l \in Labels}{\arg\max}\ PMI(l; BMU). \tag{6}$$

The PMI of two entities could take on either a positive or negative value, depending on whether they co-occur more frequently or less frequently compared to an independence assumption. The PMI measure can become zero if the two entities are independent of each other. This PMI measure is used to predict the label of a trained unit based on a hit matrix (i.e. BMU count, $\eta_u$) which stores the number of times that a unit was selected as the BMU for any input sample. The PMI uses label information from the input samples to maintain the hit matrix that is used only at the test time for accuracy calculation; as a result, the CSOM is unsupervised in training even though its test-time performance is evaluated through a supervised lens.

Note that we measure the average accuracy of SOM models based on whether the predicted SOM/CSOM unit's label matches the input sample's expected label. Unlike the L2 distance with running variance used at training time, we employed the cosine similarity at test time in order to measure the BMU for a test input sample and to determine its predicted label. Using this method, we calculated distinct accuracy measures from test samples for each task and stored these in what is known in the continual learning literature as the accuracy task matrix (of shape $T \times T$ for $T$ tasks). This helped us to calculate classical lifelong learning metrics such as average accuracy (ACC) Pham et al. (2024), backward transfer (BWT) Lopez-Paz & Ranzato (2017a), as well as the forgetting measure (FM) and learning accuracy (LA), both of which were mentioned in Yin et al. (2021). Formally, these evaluation metrics are defined as follows:

$$\text{ACC}(\uparrow) = \frac{1}{T} \Sigma_{i=1}^{T} a_{i,T} \qquad \text{(higher is better)} \tag{7}$$

$$[Rebuttal edit : \text{BWT}(\uparrow)] = \frac{1}{T-1} \Sigma_{i=1}^{T-1} a_{T,i} - a_{i,i} \qquad [\text{Rebuttal edit: (higher is better)}] \tag{8}$$

$$\text{FM}(\downarrow) = \frac{1}{T} \Sigma_{i=1}^{T} |a_{i,T} - a_i^*| \qquad \text{(lower is better)} \tag{9}$$

$$\text{LA}(\uparrow) = \frac{1}{T} \Sigma_{i=1}^{T} a_{i,i} \qquad \text{(higher is better)}. \tag{10}$$

In the above equations, $a_{i,j}$ is the accuracy for task $i$ after training on task $j$. The BWT measure helps us quantify how training on a new task $\mathcal{T}_k$ affects the accuracy of CSOM on the previously trained task $\mathcal{T}_{k-1}$. In other words, BWT helps us identify how training on a new task helped us improve performance on a previous task thus, a higher value is preferred for BWT. FM is the measure of the difference between a model's final performance and its best performance ($a_i^*$) for a task. We want forgetting (FM) to be as low as possible. LA measures the average of the current task accuracy of CSOM. A higher LA indicates better performance on an ongoing task for a model.

### 5.5 Simulation Results

In Table 1, we present our full experimental results for the three grayscale datasets. In addition, we present quantitative benchmark measurements on the CIFAR-10 dataset in Table 2, comparing to several prior, performant continual learning neural models.

| MNIST | ACC (↑) | LA (↑) | FM (↓) | BWT (↓) |
|---|---|---|---|---|
| vanilla SOM | 22.89 ± 0.02 | 46.4 ± 0.01 | 34.99 ± 0.01 | -26 ± 0.04 |
| DendSOM [a] | 17.13 ± 0.01 | 40.34 ± 0.03 | 23.33 ± 0.03 | -25.78 ± 0.04 |
| **CSOM** | **85.03 ± 4.28** | **92.26 ± 2.49** | **7.23 ± 2.42** | **-8.02 ± 2.7** |
| FMNIST | ACC (↑) | LA (↑) | FM (↓) | BWT (↓) |
| vanilla SOM | 32.07 ± 0.03 | 57.52 ± 0.03 | 37.01 ± 0.03 | -28.27 ± 0.02 |
| DendSOM | 16.62 ± 0.03 | 27.36 ± 0.03 | **11.34 ± 0.03** | **-11.92 ± 0.03** |
| **CSOM** | **75.13 ± 2.68** | **88.23 ± 0.56** | 13.11 ± 2.93 | -14.56 ± 3.25 |
| KMNIST | ACC (↑) | LA (↑) | FM (↓) | BWT (↓) |
| vanilla SOM | 23.65 ± 0.03 | 54 ± 0.05 | 35.28 ± 0.01 | -33.79 ± 0.02 |
| DendSOM | 10.95 ± 0.01 | 12.92 ± 0.01 | **2.02 ± 0** | **-2.18 ± 0.01** |
| **CSOM** | **81.602 ± 2.82** | **88.61 ± 1.6** | 7.04 ± 1.54 | -7.78 ± 1.73 |

(a) Unsupervised online (epochs = 1) class incremental training

[a][Rebuttal edit: Note that the DendSOM benchmarks are our implementation of the model, as the original source code was not publicly available]

| MNIST | ACC (↑) | LA (↑) | FM (↓) | BWT (↓) |
|---|---|---|---|---|
| vanilla SOM | 55.03 ± 0.01 | 80.07 ± 0.02 | 25.59 ± 0.02 | -30.71 ± 0.04 |
| DendSOM | 58.40 ± 0.03 | 61.37 ± 0.05 | 7.30 ± 0.02 | **-3.7 ± 0.04** |
| **CSOM** | **89.09 ± 3.32** | **93.02 ± 2.03** | **4.03 ± 1.71** | -4.92 ± 2.11 |
| FMNIST | ACC (↑) | LA (↑) | FM (↓) | BWT (↓) |
| vanilla SOM | 79.36 ± 0.06 | 89.45 ± 0.01 | 12.83 ± 0.06 | -12.61 ± 0.06 |
| DendSOM | 50.11 ± 0 | 53.13 ± 0.03 | 3.98 ± 0.03 | -3.76 ± 0.04 |
| **CSOM** | **95.61 ± 0.92** | **97.14 ± 0.4** | **1.57 ± 0.89** | **-1.92 ± 1.12** |
| KMNIST | ACC (↑) | LA (↑) | FM (↓) | BWT (↓) |
| vanilla SOM | 57.18 ± 0.01 | 78.91 ± 0.01 | 21.72 ± 0.01 | -27.16 ± 0.01 |
| DendSOM | 52.3 ± 0.02 | 54.55 ± 0.03 | **3.62 ± 0.01** | **-2.8 ± 0.02** |
| **CSOM** | **88.11 ± 1.93** | **91.97 ± 1.04** | 3.88 ± 1.04 | -4.83 ± 1.27 |

(b) Unsupervised online (epochs = 1) domain incremental learning

Table 1: Summary of (n=10 trials; mean and standard deviation of scores) results for class-incremental and domain incremental variant of vanilla SOM, DendSOM, and contSOM. For DendSOM, we created n=4 soms with unit size = 14

| Cifar-10 Model | ACC (↑) | LA (↑) | FM (↓) | BWT (↓) |
|---|---|---|---|---|
| [Rebuttal edit: Finetune] | 8.88 ± 3.07 | 37.14 ± 0.74 | 33.35 ± 2.45 | -31.40 ± 3.11 |
| LwF | 9.87 ± 0.17 | 11.37 ± 1.04 | **1.56 ± 1.31** | **-1.66 ± 1.36** |
| [Rebuttal edit: iCarl] | 28.11 ± 1.57 | 55.57 ± 1.53 | 34.32 ± 1.90 | -34.32 ± 1.90 |
| [Rebuttal edit: EWC] | 14.39 ± 2.56 | 78.11 ± 8.59 | 79.64 ± 8.55 | -79.64 ± 8.55 |
| Memo | 11.42 ± 0.48 | 96.59 ± 2.22 | 85.17 ± 1.74 | -94.64 ± 1.94 |
| [Rebuttal edit: Podnet] | 12.03 ± 1.82 | 70.17 ± 4.74 | 65.06 ± 4.57 | -64.60 ± 4.94 |
| BiC | 12.86 ± 0.72 | **96.77 ± 0.99** | 83.91 ± 1 | -93.24 ± 1.11 |
| SCALE | 29.54 ± 0.6 | 47.31 ± 3.21 | 20.23 ± 2.11 | -19.75 ± 3.52 |
| [Rebuttal edit: DER++] | 19.44 ± 0.86 | 58.83 ± 2.5 | 49.24 ± 3.49 | -49.24 ± 3.49 |
| **CSOM** | **31.3** ± 0.3 | 46.14 ± 0.48 | 14.92 ± 0.24 | -16.49 ± 0.25 |

Table 2: Summary (trials=10) of online (epochs=1) class incremental training on split-Cifar10

**The Classical SOM:** As observed in Table 1, the standard/classical SOM, or "Vanilla SOM" (which indicates that no special task-driven mechanism was integrated, thus meaning that we used the online model presented in Algorithm 1), observably forgets quite strongly across all three benchmarks/datasets. In short,

| Papers | convolution | buffer storage | Data Augmentation | single pass | class labels | task boundaries |
|---|---|---|---|---|---|---|
| Finetune | ✓ | ✗ | ✗ | ✗ | ✓ | ✗ |
| Replay | ✓ | ✓ | ✗ | ✗ | ✓ | ✓ |
| LwF Li & Hoiem (2018) | ✓ | ✗ | ✗ | ✗ | ✓ | ✓ |
| iCarl Rebuffi et al. (2017) | ✓ | ✓ | ✗ | ✗ | ✓ | ✓ |
| EWC Kirkpatrick et al. (2017) | ✗ | ✗ | ✗ | ✗ | ✓ | ✓ |
| Memo Zhou et al. (2023b) | ✓ | ✓ | ✗ | ✗ | ✓ | ✓ |
| Podnet Douillard et al. (2020) | {✓, ✗} | ✓ | ✗ | ✗ | ✓ | ✓ |
| BiC Wu et al. (2019) | ✓ | ✓ | ✗ | ✗ | ✓ | ✓ |
| SCALE Yu et al. (2022) | ✓ | ✓ | ✓ | ✓ | ✗ | ✗ |
| [Rebuttal edit: DER++] Boschini et al. (2022) | ✓ | ✓ | ✓ | ✗ | ✗ | ✗ |
| CSOM | ✗ | ✗ | ✗ | ✓ | ✗ | ✗ |

Table 3: Training schemes followed by all approaches

our results empirically confirm that, indeed, the standard SOM forgets information despite the potential (for reduced neural cross-talk) offered by its competitive internal activities. We present visual samples of the memories acquired by our SOM model(s) in Figure 4, which qualitatively demonstrates the types of memories these specific models acquire during learning – for datasets such as Split-MNIST, the SOM only has stored in its internal memories prototypes/templates that match the last few digits/tasks of the underlying task sequence, i.e., largely the digit "8" and "9".

**The CSOM:**  The conventional distance metrics, such as euclidean distance or cosine similarity, often identify one of the previously trained units as the BMU for the current incoming task samples due to the semantic overlap present among different task samples in datasets like MNIST (as explained in section 5.1). This means that the images from different classes in MNIST-like datasets appear to be quite similar due to the common black background and white foreground. This increases the probability that an already trained SOM unit is selected as BMU for an input sample from a new class instead of an untrained unit being selected as BMU. Thus, a vanilla SOM fails to use its full capacity and ultimately does not train on all the available units in a continual learning setting. The distance function shown in the TRAIN() subroutine of algorithm 2, divides the squared terms of the L2 distance by the running standard deviation of units obtained from $\mathbf{M}^\sigma$. This helps to preserve the trained units' memories (with respect to previously encountered classes) and successfully allocates the untrained units in the SOM to new incoming tasks. In other words, such a distance metric for identifying BMU helps to balance the stability and plasticity of the neural system. More importantly, it eliminates the need for a task vector (refer to Table 3), which sets up context for the incoming tasks in most modern-day continual learning models. As a result, the CSOM can learn in a streaming manner without needing task boundaries which makes it a powerful, unsupervised lifelong learning system.

**Uniqueness of CSOM:**  [Rebuttal edit: Experimentally, we noticed that the Gaussian function used to scale the synaptic weight updates in a neighborhood was liable to cause leaky weight updates in the units that were very far away from the BMU in topology. This caused an explosion of weight update in the available untrained units, thus undesirably increasing the size of cluster for current incoming task. To control this behavior, we enforced a hard bound over the synaptic weight update in the neighborhood using the current $\sigma_u$ value of the BMU. For this we create a binary mask as shown in step 6 of UPDATE() subroutine in Algorithm 3. Since the goal of CSOM is to have linearly separable trained clusters of streaming input data, it is required that the radius and learning rate also remain independent for different classes/tasks. A global copy of radius and learning rate set to high value initially leads to larger clusters for initial classes/tasks passed in the streaming setting and smaller clusters for the ones appearing later. This creates imbalanced clustering in the SOM as supported by our empirical trials. With each unit having its own copy of radius and learning rate in CSOM, every class/task gets the opportunity to form a cluster according to variance of its respective input samples. The running standard deviation used in the distance metric shown in line 5 of Algorithm 2 is decayed according to the number of times a unit has been selected as BMU. It reduces the chances

of an already well trained unit to get selected again as BMU for similar looking sample but belonging to different class (eg, 3 and 8). An untrained or less trained neighboring unit will have relatively higher running standard deviation thus giving a lower distance value from the input sample. This promotes formation of cluster in the empty space of CSOM creating better quality of trained clusters towards the end as seen in Figure 4 and 5. Also, this follows the principle of competitive learning among the CSOM units leading to better performance and faster convergence than vanilla SOM. This distance function also makes CSOM immune to sudden changes in the distribution of input without task boundary heuristic. Further details on hyperparameter tuning is provided in the appendix.]

[Rebuttal edit: In Table 1 we show benchmark comparison of CSOM with other variants on grayscale images datasets. DendSOM Pinitas et al. (2021a), is a variant of SOM designed for continual learning setting but fails to perform well in online unsupervised setting as seen from its result values. In Table 2, we present a comparison of model performance measurements on the split-CIFAR10 benchmark. We used Mammoth Boschini et al. (2023) to obtain benchmarks for EWC, iCarl and DER++ whereas PyCIL Zhou et al. (2023a) to obtain all others for standard continual learning models except SCALE Yu et al. (2022).] For them, we kept the same setting of total memory size=2000 and memory per class=20. These were the standard parameter settings that Zhou et al. (2023a) used to obtain their baseline results. We conducted 10 experimental trials for all the benchmarks and, after every trial, we constructed a task matrix in order to calculate the ACC, LA, FM, and BWT values for any model. [Rebuttal edit: DendSOM, SCALE and DER++ follow similar setting as CSOM in the algorithm]. Among all the benchmarks, Yu et al. (2022) particularly had the exact same setting (shown in Table 3) as ours and had state-of-the-art (SOTA) ACC values in a semi-supervised learning setting. Although it had marginally higher mean LA value, their standard deviation was almost 7 times higher than our CSOM after 10 trials. Due to the online setting of single epoch and very low buffer storage, few algorithms experience more catastrophic forgetting thus giving us close to 0 standard deviation in their evaluation over ($n = 10$) trials. Except for EWC Kirkpatrick et al. (2017), all of the models used convolution and most of them used a storage buffer (memory). Nevertheless, our CSOM model yielded the best average accuracy among all others without maintaining any buffer storage or requiring convolution to obtain better performance, despite operating in an online setting (epochs=1, each input sample processed once). [Rebuttal edit: The codebase for SSOCL, Imai et al. (2024) and R2R Mandalika et al. (2025) is not publicly available and hence we could not obtain benchmarks on them.]

As shown in Figure 4, as a result of the neural clusters formed, our model appeared to acquire a good latent representation of every input task. This means that there is a good amount of variance among the trained units allocated for every task. However, despite our good experimental results, we could not obtain an equal number of trained units per class. Moreover, some neuronal units (especially those at the line of separability of clusters) have fuzzy representations. This may occur if a neuronal unit receives a weight update that corresponds to multiple different classes.

Nevertheless, as indicated in Table 1 and 2, **the CSOM achieves the best performance compared to all of the other models/variants of SOM on all four benchmarks/datasets and successfully beats the SOTA model on split-CIFAR10 dataset**. This empirical result indicates that our CSOM framework, leveraging a competitive learning scheme, can yield the potentially best memory retention (or greatest reduction in forgetting) when processing classes incrementally from a data stream. The visualization of the samples synthesized for this particular model – see Figure 4 – also corroborate this result qualitatively; the samples, in this case, look the clearest and the model's internal units seem to represent most of the individual classes/tasks (including those presented at the start of the task sequence).

**Discussion:** The nature of the distance function and the hyperparameter decay method used in all the SOM models induces an inherent bias towards the initially encountered classes in the class incremental setting. Specifically, it assigns a larger cluster or more trained units in the SOM to initial classes compared to classes that appear later in the incremental order. Although the proposed CSOM produced the best results out of all of the model variants, further improvement is certainly possible in order to obtain clearer/crisper representations of input samples in a trained model final output of a trained SOM (this will be the subject of future work). By augmenting the SOM's internal dynamics with additional synaptic parameters that maintained the running variance for all of its neuronal units, we show that the CSOM can further be used as a useful generative model, facilitating a natural internal memory replay mechanism that could further

aid in memory retention in more complex continual learning setups. Notably, this could be an important component to improve the performance of memory-augmented neural networks Das et al. (1992); Graves et al. (2014); Joulin & Mikolov (2015); Mali et al. (2020); Stogin et al. (2024).

# 6 Conclusions

In this paper, we investigated catastrophic forgetting – a phenomenon where a neural-based intelligent agent forgets the information/knowledge that it acquired on previous datasets/tasks whenever it starts processing a new dataset/task – in the context of continual unsupervised learning utilizing the classical self-organizing map (SOM). Specifically, we designed an adaptation of the classical model to this setting and empirically found that it exhibited severe interference and low memory retention. In light of this, we proposed a novel generalization of the model – the continual SOM (CSOM) – which we theoretically and experimentally demonstrated, across several data benchmarks, that it exhibited vastly improved memory retention ability notably through the use of running variance and decay mechanisms, locally embedded into each neuronal unit. Future work will include an examination of the integration of the cross-task memory retention ability of our model into supervised and semi-supervised neural systems on additional, larger-scale data problems, as well as further improving the quality of the CSOM's internally acquired neural prototypes through additional extensions/generalizations of its local unit-variance parameters.

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

## A  Appendix

In addition to Algorithm 2, we provide its pseudocode in Algorithm 3.

**Revisiting the hyperparameter setting:**

Table 4 indicates key notations, symbols and abbreviations used in this paper. Figures 6, 7, 8 show relation between parameters and their consequent metrics obtained using Fashion MNIST dataset. We empirically found that increasing learning rates leads to faster adaptation of CSOM to the innput data but may have chance of instability or overfitting. There may be semantic overlap due to high learning rate values for CSOM

---

**Algorithm 3** [Rebuttal edit: pseudocode of CSOM]

---

**Require:** data sample $\mathbf{x}_i^{(k)}(t)$, SOM weight matrix $\mathbf{M}$, running variance matrix $\mathbf{M}^{\omega^2} \in \mathcal{R}^{D \times H}$, radius $\sigma_v, \sigma_V$, learning rate $\lambda_h$, BMU count for a unit ($\eta_u$ is count for unit $u$)

  1: **Note:** HTile($\mathbf{x}, H$) tiles vector $\mathbf{v}$ $H$ times horizontally; VTile($\mathbf{x}, H$) tiles vector $\mathbf{x}$ $H$ times vertically,

  2:      $d(a, b; \mathcal{G})$ is a distance over topology $\mathcal{G}$, where $a$ maps to $(a_i, a_j)$ (2D coordinate) and $b$ maps to $(b_i, b_j)$

  3: **function** UPDATE($\mathbf{X}_i, u, V, \sigma_u^{(t)}, \lambda_u^{(t)}, \mathbf{M}, \mathbf{M}^{\omega^2}, \mathcal{G}$)

  4:      $\delta = (2 * \sigma_u^2)^{-1}$

  5:      $\tau_1 = -\delta^{-1} * \log\left(\frac{10^{-8}}{\lambda_u}\right)$

  6:      $\mathbf{o}_{1,v_j} = \begin{cases} 1 & d(v_j, u; \mathcal{G}) < \sigma_u \\ 0 & \text{otherwise} \end{cases}$          ▷ $v_j$ is any non-BMU neuron, $\mathbf{o} \in \{0,1\}^{1 \times H}$ (masking vector)

  7:      $\mathbf{s}_{1,j} = \begin{cases} \lambda_u & j = u \\ \lambda_v & j = v_j \end{cases}$          ▷ $\mathbf{s} \in \mathcal{R}^{1 \times H}$ learning rate vector

  8:      $\phi = \text{VTile}\left(\{\mathbf{o}_{v_j} \mathbf{s}_{v_j} \exp\left[-d(u, j; \mathcal{G})\delta\right], \text{ for } j = 1, 2, ..., H\}, D\right)$      ▷ $\phi \in \mathcal{R}^{D \times H}$

  9:      $\mathbf{M} = \mathbf{M} + \phi \odot (\mathbf{X}_i - \mathbf{M})$      ▷ CSOM update step

10:      $\eta_u = \eta_u + 1$

11:      $\lambda_\omega = \text{VTile}\left(\{(\lambda_\omega^0 - 0.5) + (1 + \exp\left(-d(u, j; \mathcal{G})/\tau_1\right))^{-1}, \text{ for } j = 1, 2, ..., H\}, D\right)$      ▷ $\lambda_\omega \in \mathcal{R}^{D \times H}$

12:      $\lambda_\omega = \lambda_\omega \odot \mathbf{O} + (1 - \mathbf{O})$, where $\mathbf{O} = \text{VTile}(\mathbf{o}, D)$      ▷ scaling factor for updating running variance

13:      $\mathbf{M}^{\omega^2} = \lambda_\omega \mathbf{M}^{\omega^2} + (1 - \lambda_\omega)(\mathbf{X}_i - \mathbf{M})^2$      ▷ update running variance of all neurons

14:      [Rebuttal edit: $\sigma_u^{(t)} = \sigma_u^{(t-1)} \exp\left(-\eta_u/\tau_\sigma\right)$ ]      ▷ $\tau_\sigma \leftarrow$ constant

15:      [Rebuttal edit: $\lambda_u^{(t)} = \lambda_u^{(t-1)} \exp\left(-\eta_u/\tau_\lambda\right)$ ]      ▷ $\tau_\lambda \leftarrow$ constant

16: **function** TRAIN($\mathbf{x}_i^{(k)}(t), \mathbf{M}, \mathbf{M}^{\omega^2}, \mathcal{G}, t$)

17:      $\mathbf{X}_i \leftarrow \text{HTile}(\mathbf{x}_i^{(k)}(t), H)$      ▷ tile/repeat $\mathbf{x}_i^{(t)}$ $H$ times horizontally to create $\mathbf{x}_i^{(k)}(t) = \mathbf{X}_i \in \mathcal{R}^{D \times H}$

18:      $\boldsymbol{\Delta}_\omega = (\mathbf{X}_i - \mathbf{M})^2 / \mathbf{M}^\omega$      ▷ distance calculation specific to CSOM , $\mathbf{M}^\omega = \sqrt{\mathbf{M}^{\omega^2}}$

19:      $\mathbf{d} = \Delta_\omega[i, j]$      ▷ $i \in \{1, 2, ..K\}$ and $j \in \{1, 2, ..L\}$

20:      $u = \arg\min_{h \in H} \mathbf{d}$      ▷ BMU calculation

21:      UPDATE($X_i, u, V, \sigma_u^{(t)}, \lambda_u^{(t)}, \mathbf{M}, \mathbf{M}^{\omega^2}, \mathcal{G}$)      ▷ $V = H \setminus u$ (i.e., $V$ is set of all non-BMU indices)

---

units or the resultant cluster of a class may not capture good variance of input samples in CSOM. Low learning rate values lead to slower convergence. CSOM may be more stable in this case but may also lead to underfitting if the learning rate value is too low. Higher radius values of CSOM units leads to broader neighborhood influence that is bigger clusters of classes from input data distribution. Lower radius values on the other hand lead to lesser interference on the line of separability between clusters. A grid search with CSOM over all datasets indicated that initial running variance has to be just a little higher than the variance of the input distribution to get distinct clusters in the trained model. At the same time, it is important to keep the initial radius value just $> 1$ but not higher than 2 because that gives more spread for weight update to every BMU thus failing to capture a good compressed distribution of input samples in the finally trained CSOM. Since the distribution of $\mathbf{M}^{\omega^2}$ is just higher than input distribution and its updates are tied with the weight update of $\mathbf{M}$, the initial $\lambda_\omega^0$ also has to be higher than $\lambda_0$. This can be seen from their choices mentioned in Tables 5, 6, 8, 10, 12, 16, 18, 20, and 22. The scaling factors $\tau_\sigma$ and $\tau_\lambda$ contribute less to learning compared to $\sigma$ and $\lambda$ as evident from the correlation matrix in Figure 8. Experimentally we have found that values in the range of 7-9 work well for $\tau_\sigma$ and 40-45 for $\tau_\lambda$.

**scaling factor $\lambda_\omega$ in CSOM** As shown in pseudocode 3, $sigmoid(.)$ gets the input of $-unit\_distance/\tau_1$. $\tau_1$ is calculated in runtime at line 5 in pseudocode 3. If we assume the initial parameter setting mentioned in Table 5 and substitute them in the equation for $\lambda_\omega$ then we get,

| Item | Explanation |
|------|-------------|
| $\mathcal{G} = (K \times L)$ | Network Topology (dimensions of the SOM) |
| | **NOTE:** $K = L$ for CSOM |
| $D$ | unit/neuron size and size of individual input sample |
| $\mathbf{M} \in \mathcal{R}^{D \times H}$ | synaptic matrix of SOM |
| $u$ | Best Matching Unit (BMU) |
| $v_j$ | a non-BMU at location $j$ in CSOM |
| $V = \{v_j \mid 0 < j < K^2\}$ | all non-BMU units in CSOM |
| $H = \{u\} \cup V$ | all units in the SOM |
| $\sigma_0$ | initial radius for all neurons |
| $\lambda_0$ | initial learning rate all neurons |
| $\sigma_u$ and $\sigma_v$ | radius of BMU and radius of non-BMU neuron |
| $\lambda_u$ and $\lambda_v$ | learning rate of BMU and learning rate of non-BMU neuron |
| $\omega^2$ | running variance of a neuron |
| $\mathbf{M}^{\omega^2} \in \mathcal{R}^{D \times H}$ | matrix of running variance of all neurons in SOM |
| $\sigma_h = \{\sigma_u\} \cup \{\sigma_{v_j}\} \in \mathcal{R}^{1 \times H}$ | radius of all neurons for weight update |
| $\lambda_h = \{\lambda_u\} \cup \{\lambda_{v_j}\} \in \mathcal{R}^{1 \times H}$ | learning rate for all neurons used in the weight update steps of CSOM |
| $\lambda_\omega^0$ | initial scaling/update factor for updating the running variance of every unit |
| $\lambda_\omega$ | adjusted scaling/update factor for updating the running variance of every unit |
| $\tau_\sigma$ | time constant for radius |
| $\tau_\lambda$ | time constant for learning rate |
| p | patch size (DendSOM) |
| s | stride length (DendSOM) |

Table 4: Notations, Symbols, abbreviations used in this paper

| Model | $(K = L)$ | $\sigma_0$ | $\lambda_0$ | $\omega_0^2$ | $\lambda_\omega^0$ |
|-------|-----------|-----------|-----------|-----------|-----------|
| Vanilla SOM | [15, 20, 20] | 0.6 | 0.07 | 1 | 0 |
| DendSOM | [10, 10, 10] | 1.5 | 0.07 | 0 | 0 |
| CSOM | [15/20, 25, 35] | 1.5 | 0.07 | 0.5 | 0.9 |

Table 5: [Rebuttal edit: Parameter setting for the architectures and their training]

$$\lambda_\omega = (0.9 - 0.5) \; + \; \cfrac{1}{\left(1 \; + \; \exp\left(-\cfrac{x}{2 \cdot (1.2)^2 \cdot \log\left(\frac{10^{-8}}{0.07}\right)}\right)\right)}$$

We plot this equation on a graph as shown in Figure 9 to prove that the overall value never exceeds 1.

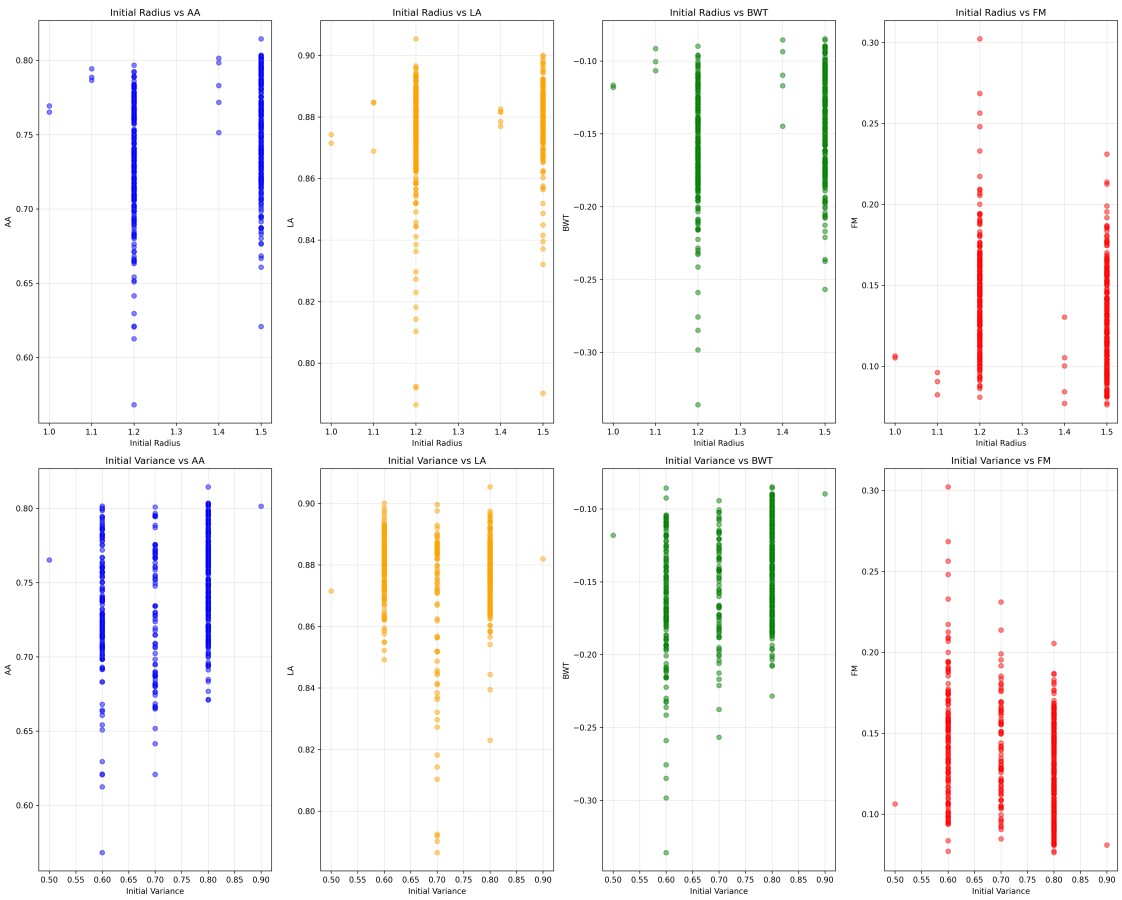

Figure 6: [Rebuttal edit: parameter sensitivity of model trained on fashion MNIST measured in relation with result metrics]

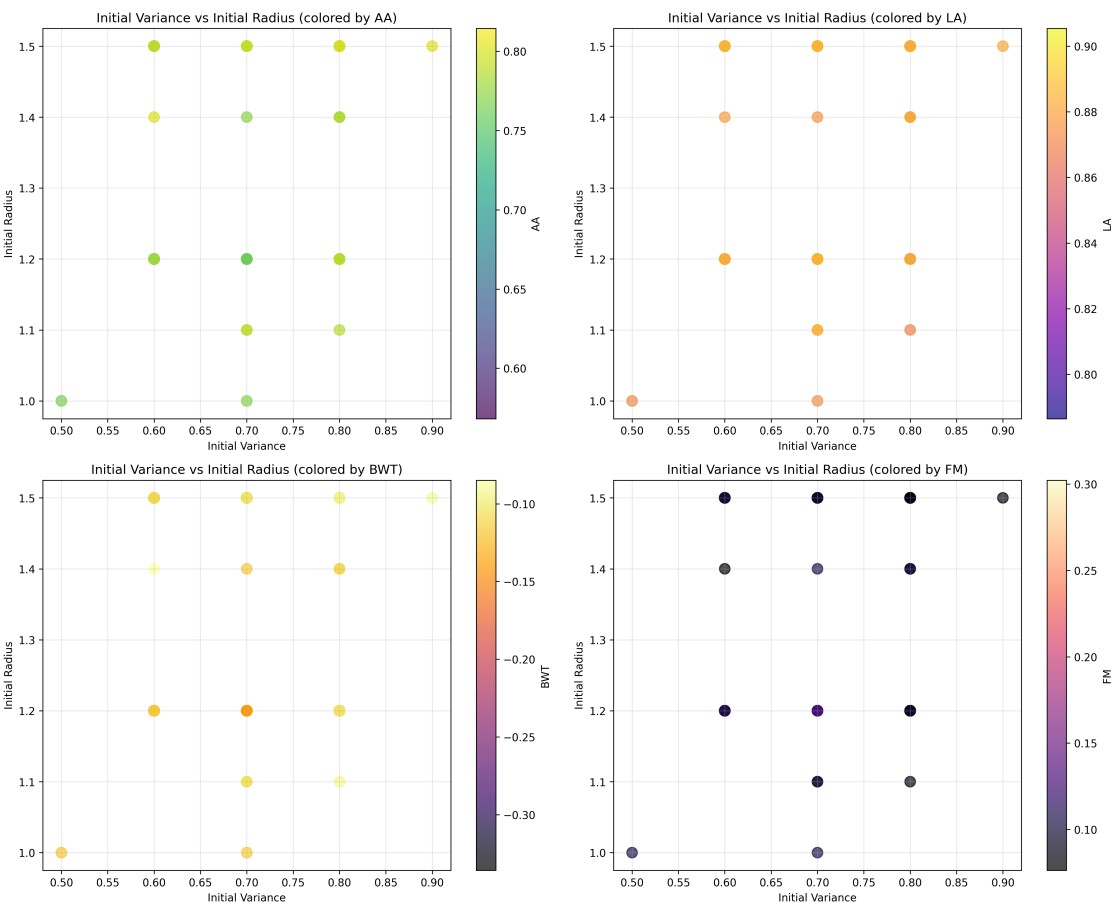

Figure 7: [Rebuttal edit: Relation between metrics and combination of parameters for CSOM trained on fashion MNISt]

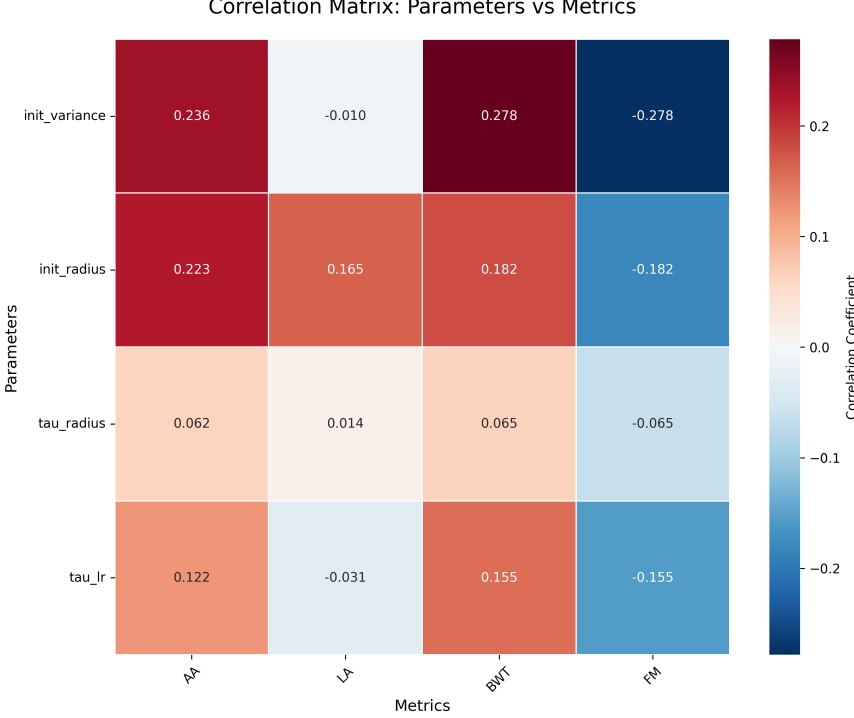

Figure 8: [Rebuttal edit: Correlation matrix of parameters and metrics for fashion MNIST]

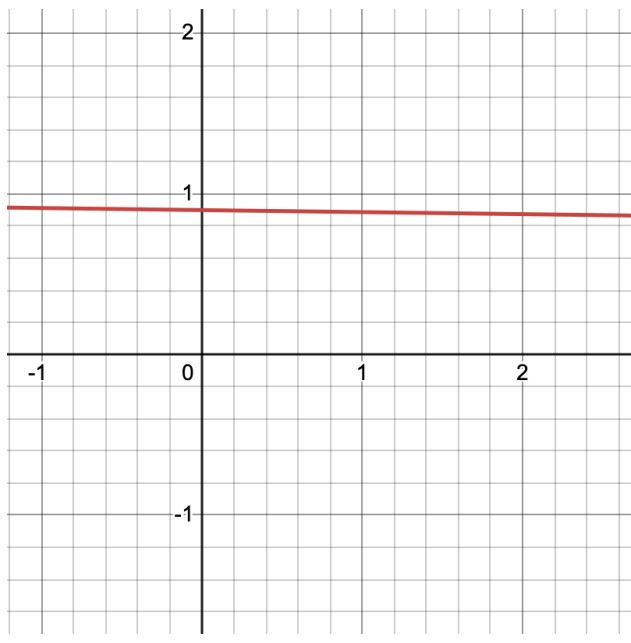

Figure 9: decay of $\lambda_\omega$ as per line 11 in Algorithm 3

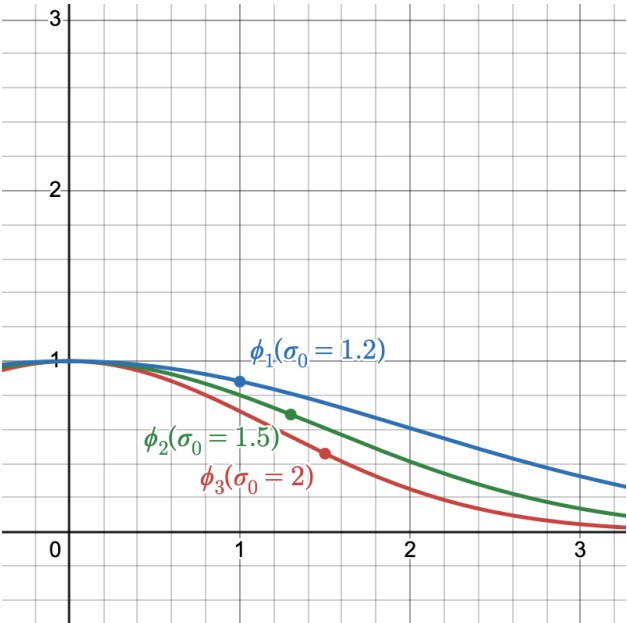

Figure 10: [Rebuttal edit: relation between the initial radius and the neighborhood function (is independent of running variance) for weight update]

## A.1 Neighborhood function

The neighborhood function is generally a gaussian curve (Figure 10) which dictates the amount of weight update to be performed for units in the neighborhood of the best matching unit (including itself) for an input sample. The $\delta$ mentioned in line 5 of Algorithm 2, is used as a denominator for the calculation of neighborhood function in line 8. The neighborhood function is always independent of the running variance in case of CSOM.

## A.2 Class Incremental Learning

This section contains information about class incremental settings where each task size had exactly one class in it.

### A.2.1 MNIST

**Hyperparameters:** Table 6 contains the empirically obtained initial hyperparameters for obtaining the best possible results as described in Table 1a.

| Model | $H$ | $D$ | $\sigma_0$ | $\lambda_0$ | $\omega_0^2$ | $\lambda_\omega^0$ | $\tau_\sigma$ | $\tau_\lambda$ | p | s |
|---|---|---|---|---|---|---|---|---|---|---|
| Vanilla SOM | $15 \times 15$ | $28 \times 28$ | 0.6 | 0.07 | 1 | 0.9 | 8 | 45 | - | - |
| DendSOM | $8 \times 8$ | $7 \times [14 \times 14]$ | 4 | 0.95 | 2 | 0.005 | - | - | 10 | 3 |
| CSOM | $15 \times 15$ | $28 \times 28$ | 1.5 | 0.07 | 0.5 | 0.9 | 8 | 45 | - | - |

Table 6: Hyperparameters for Class Incremental setting on MNIST

**Task matrices:** We performed 10 trials of CSOM trained on MNIST. Table 7 indicates the resultant accuracy values,

| | | | | | | | | | |
|---|---|---|---|---|---|---|---|---|---|
| $100.0 \pm 0.0$ | $0.0 \pm 0.0$ | $0.0 \pm 0.0$ | $0.0 \pm 0.0$ | $0.0 \pm 0.0$ | $0.0 \pm 0.0$ | $0.0 \pm 0.0$ | $0.0 \pm 0.0$ | $0.0 \pm 0.0$ | $0.0 \pm 0.0$ |
| $99.77 \pm 0.12$ | $99.89 \pm 0.14$ | $0.0 \pm 0.0$ | $0.0 \pm 0.0$ | $0.0 \pm 0.0$ | $0.0 \pm 0.0$ | $0.0 \pm 0.0$ | $0.0 \pm 0.0$ | $0.0 \pm 0.0$ | $0.0 \pm 0.0$ |
| $99.17 \pm 0.4$ | $99.41 \pm 0.31$ | $96.91 \pm 1.2$ | $0.0 \pm 0.0$ | $0.0 \pm 0.0$ | $0.0 \pm 0.0$ | $0.0 \pm 0.0$ | $0.0 \pm 0.0$ | $0.0 \pm 0.0$ | $0.0 \pm 0.0$ |
| $98.83 \pm 0.54$ | $99.23 \pm 0.31$ | $93.31 \pm 2.98$ | $96.01 \pm 1.41$ | $0.0 \pm 0.0$ | $0.0 \pm 0.0$ | $0.0 \pm 0.0$ | $0.0 \pm 0.0$ | $0.0 \pm 0.0$ | $0.0 \pm 0.0$ |
| $98.78 \pm 0.52$ | $99.22 \pm 0.32$ | $92.26 \pm 3.38$ | $94.61 \pm 1.8$ | $96.34 \pm 1.02$ | $0.0 \pm 0.0$ | $0.0 \pm 0.0$ | $0.0 \pm 0.0$ | $0.0 \pm 0.0$ | $0.0 \pm 0.0$ |
| $96.78 \pm 3.13$ | $99.07 \pm 0.43$ | $92.12 \pm 3.39$ | $84.99 \pm 9.97$ | $96.05 \pm 1.02$ | $88.98 \pm 5.21$ | $0.0 \pm 0.0$ | $0.0 \pm 0.0$ | $0.0 \pm 0.0$ | $0.0 \pm 0.0$ |
| $95.96 \pm 3.0$ | $98.5 \pm 1.22$ | $89.5 \pm 6.05$ | $84.52 \pm 10.0$ | $95.39 \pm 1.19$ | $86.19 \pm 6.37$ | $94.8 \pm 2.46$ | $0.0 \pm 0.0$ | $0.0 \pm 0.0$ | $0.0 \pm 0.0$ |
| $95.93 \pm 2.99$ | $98.5 \pm 1.22$ | $88.33 \pm 6.72$ | $83.39 \pm 10.11$ | $93.3 \pm 2.96$ | $85.32 \pm 6.34$ | $94.75 \pm 2.48$ | $92.54 \pm 1.59$ | $0.0 \pm 0.0$ | $0.0 \pm 0.0$ |
| $95.93 \pm 3.03$ | $98.41 \pm 1.19$ | $87.09 \pm 6.55$ | $81.73 \pm 9.47$ | $93.05 \pm 3.07$ | $80.09 \pm 9.43$ | $93.6 \pm 4.35$ | $92.42 \pm 1.6$ | $84.72 \pm 2.25$ | $0.0 \pm 0.0$ |
| $95.89 \pm 3.02$ | $98.41 \pm 1.19$ | $87.14 \pm 6.55$ | $81.39 \pm 9.33$ | $79.17 \pm 8.47$ | $79.53 \pm 9.35$ | $93.36 \pm 4.28$ | $80.64 \pm 7.77$ | $82.43 \pm 3.17$ | $72.37 \pm 14.48$ |

Table 7: mean and standard deviation of accuracies in a task matrix

## A.2.2 KMNIST

**Hyperparameters:** Table 8 shows the hyperparameters we used for obtaining the results in Table 1a on KMNIST dataset.

| Model | $H$ | $D$ | $\sigma_0$ | $\lambda_0$ | $\omega_0^2$ | $\lambda_\omega^0$ | $\tau_\sigma$ | $\tau_\lambda$ | p | s |
|---|---|---|---|---|---|---|---|---|---|---|
| Vanilla SOM | $20 \times 20$ | $28 \times 28$ | 0.6 | 0.07 | 1 | 0.9 | 8 | 45 | - | - |
| DendSOM | $10 \times 10$ | $7 \times [14 \times 14]$ | 6 | 0.95 | 2 | 0.005 | - | - | 4 | 2 |
| CSOM | $35 \times 35$ | $28 \times 28$ | 1.5 | 0.07 | 0.5 | 0.9 | 8 | 45 | - | - |

Table 8: Hyperparameters for Class Incremental setting on KMNIST

**Task matrices:** We performed 10 trials of CSOM trained on KMNIST. Table 9 indicates the resultant accuracy values,

| | | | | | | | | | |
|---|---|---|---|---|---|---|---|---|---|
| $100.0 \pm 0.0$ | $0.0 \pm 0.0$ | $0.0 \pm 0.0$ | $0.0 \pm 0.0$ | $0.0 \pm 0.0$ | $0.0 \pm 0.0$ | $0.0 \pm 0.0$ | $0.0 \pm 0.0$ | $0.0 \pm 0.0$ | $0.0 \pm 0.0$ |
| $98.08 \pm 0.85$ | $98.72 \pm 0.71$ | $0.0 \pm 0.0$ | $0.0 \pm 0.0$ | $0.0 \pm 0.0$ | $0.0 \pm 0.0$ | $0.0 \pm 0.0$ | $0.0 \pm 0.0$ | $0.0 \pm 0.0$ | $0.0 \pm 0.0$ |
| $97.86 \pm 0.71$ | $88.0 \pm 2.71$ | $93.25 \pm 1.14$ | $0.0 \pm 0.0$ | $0.0 \pm 0.0$ | $0.0 \pm 0.0$ | $0.0 \pm 0.0$ | $0.0 \pm 0.0$ | $0.0 \pm 0.0$ | $0.0 \pm 0.0$ |
| $97.44 \pm 0.82$ | $88.06 \pm 2.81$ | $87.59 \pm 1.93$ | $94.2 \pm 1.67$ | $0.0 \pm 0.0$ | $0.0 \pm 0.0$ | $0.0 \pm 0.0$ | $0.0 \pm 0.0$ | $0.0 \pm 0.0$ | $0.0 \pm 0.0$ |
| $93.98 \pm 1.22$ | $86.21 \pm 3.18$ | $86.63 \pm 1.83$ | $93.53 \pm 1.66$ | $82.79 \pm 1.95$ | $0.0 \pm 0.0$ | $0.0 \pm 0.0$ | $0.0 \pm 0.0$ | $0.0 \pm 0.0$ | $0.0 \pm 0.0$ |
| $89.3 \pm 3.37$ | $85.93 \pm 3.2$ | $84.0 \pm 3.27$ | $91.73 \pm 2.03$ | $81.15 \pm 3.03$ | $82.6 \pm 3.57$ | $0.0 \pm 0.0$ | $0.0 \pm 0.0$ | $0.0 \pm 0.0$ | $0.0 \pm 0.0$ |
| $88.97 \pm 3.34$ | $83.55 \pm 3.24$ | $79.92 \pm 3.71$ | $91.27 \pm 2.14$ | $79.93 \pm 3.2$ | $80.48 \pm 3.16$ | $85.91 \pm 3.51$ | $0.0 \pm 0.0$ | $0.0 \pm 0.0$ | $0.0 \pm 0.0$ |
| $86.32 \pm 3.14$ | $83.37 \pm 3.24$ | $79.42 \pm 3.74$ | $91.16 \pm 2.19$ | $79.43 \pm 3.17$ | $80.33 \pm 3.13$ | $85.68 \pm 3.49$ | $89.19 \pm 1.55$ | $0.0 \pm 0.0$ | $0.0 \pm 0.0$ |
| $83.86 \pm 3.45$ | $81.31 \pm 3.21$ | $77.98 \pm 3.72$ | $90.45 \pm 2.68$ | $77.59 \pm 3.58$ | $79.34 \pm 3.38$ | $83.33 \pm 4.9$ | $87.31 \pm 2.63$ | $80.75 \pm 4.35$ | $0.0 \pm 0.0$ |
| $83.75 \pm 3.41$ | $80.35 \pm 3.18$ | $77.21 \pm 3.98$ | $90.39 \pm 2.72$ | $75.72 \pm 4.39$ | $79.28 \pm 3.41$ | $83.25 \pm 4.92$ | $86.62 \pm 2.2$ | $80.79 \pm 4.36$ | $78.66 \pm 4.79$ |

Table 9: mean and standard deviation of accuracies in a task matrix

## A.2.3 Fashion-MNIST

**Hyperparameters:** Table 10 indicates the hyperparameters used for obtaining results in Table 1a on Fashion-MNIST dataset.

## A.2.4 CIFAR10

**Hyperparameters:** Table 12 indicates the hyperparameters used to obtain results in Table 2 the on split-CIFAR10 dataset.

Table 13 indicates hyperparameters set while implementing models from Zhou et al. (2023a) in a Class Incremental Online Learning setting.

Apart from the hyperparameters mentioned in Tables 13 and 14, the rest of the hyperparameters were unchanged in their publicly available code. We added an evaluation code for obtaining task matrices for all the models in Table 2.

**Task matrices:** We performed 8 trials of CSOM trained on CIFAR-10. Following resultant accuracies values.

| Model | $H$ | $D$ | $\sigma_0$ | $\lambda_0$ | $\omega_0^2$ | $\lambda_\omega^0$ | $\tau_\sigma$ | $\tau_\lambda$ | p | s |
|---|---|---|---|---|---|---|---|---|---|---|
| Vanilla SOM | $20 \times 20$ | $28 \times 28$ | 0.6 | 0.07 | 1 | 0.9 | 8 | 45 | - | - |
| DendSOM | $10 \times 10$ | $7 \times [14 \times 14]$ | 5 | 0.95 | 2 | 0.005 | - | - | 8 | 4 |
| CSOM | $25 \times 25$ | $28 \times 28$ | 1.5 | 0.07 | 0.5 | 0.9 | 8 | 45 | - | - |

Table 10: Hyperparameters for Class Incremental setting on Fashion-MNIST

| | | | | | | | | | |
|---|---|---|---|---|---|---|---|---|---|
| 100.0 ± 0.0 | 0.0 ± 0.0 | 0.0 ± 0.0 | 0.0 ± 0.0 | 0.0 ± 0.0 | 0.0 ± 0.0 | 0.0 ± 0.0 | 0.0 ± 0.0 | 0.0 ± 0.0 | 0.0 ± 0.0 |
| 97.39 ± 1.49 | 97.7 ± 0.59 | 0.0 ± 0.0 | 0.0 ± 0.0 | 0.0 ± 0.0 | 0.0 ± 0.0 | 0.0 ± 0.0 | 0.0 ± 0.0 | 0.0 ± 0.0 | 0.0 ± 0.0 |
| 92.9 ± 1.91 | 96.72 ± 0.82 | 94.32 ± 0.71 | 0.0 ± 0.0 | 0.0 ± 0.0 | 0.0 ± 0.0 | 0.0 ± 0.0 | 0.0 ± 0.0 | 0.0 ± 0.0 | 0.0 ± 0.0 |
| 85.83 ± 3.25 | 88.44 ± 6.73 | 93.67 ± 0.72 | 89.49 ± 3.6 | 0.0 ± 0.0 | 0.0 ± 0.0 | 0.0 ± 0.0 | 0.0 ± 0.0 | 0.0 ± 0.0 | 0.0 ± 0.0 |
| 85.71 ± 2.99 | 88.34 ± 6.68 | 73.42 ± 4.09 | 83.11 ± 4.61 | 76.2 ± 3.7 | 0.0 ± 0.0 | 0.0 ± 0.0 | 0.0 ± 0.0 | 0.0 ± 0.0 | 0.0 ± 0.0 |
| 85.64 ± 3.03 | 88.34 ± 6.68 | 73.49 ± 4.08 | 83.07 ± 4.58 | 75.93 ± 3.79 | 99.76 ± 0.11 | 0.0 ± 0.0 | 0.0 ± 0.0 | 0.0 ± 0.0 | 0.0 ± 0.0 |
| 77.5 ± 3.31 | 88.34 ± 6.69 | 68.65 ± 3.46 | 82.88 ± 4.54 | 72.45 ± 3.56 | 99.6 ± 0.39 | 41.62 ± 2.74 | 0.0 ± 0.0 | 0.0 ± 0.0 | 0.0 ± 0.0 |
| 77.51 ± 3.32 | 88.34 ± 6.69 | 68.65 ± 3.48 | 82.88 ± 4.54 | 72.45 ± 3.56 | 66.87 ± 13.25 | 41.55 ± 2.73 | 95.77 ± 3.74 | 0.0 ± 0.0 | 0.0 ± 0.0 |
| 77.24 ± 3.24 | 88.33 ± 6.68 | 68.57 ± 3.48 | 82.62 ± 4.61 | 72.11 ± 3.42 | 66.83 ± 13.1 | 39.95 ± 2.75 | 90.12 ± 9.74 | 94.84 ± 0.94 | 0.0 ± 0.0 |
| 77.23 ± 3.21 | 88.34 ± 6.66 | 68.53 ± 3.47 | 82.6 ± 4.59 | 72.08 ± 3.4 | 58.84 ± 14.3 | 39.87 ± 2.69 | 76.89 ± 11.12 | 94.28 ± 1.19 | 92.65 ± 2.89 |

Table 11: mean and standard deviation of accuracies in a task matrix

| Model | $H$ | $D$ | $\sigma_0$ | $\lambda_0$ | $\omega_0^2$ | $\lambda_\omega^0$ | $\tau_\sigma$ | $\tau_\lambda$ |
|---|---|---|---|---|---|---|---|---|
| CSOM | $100 \times 100$ | $32 \times 32$ | 1.5 | 0.2 | 0.6 | 0.9 | 6 | 45 |

Table 12: Hyperparameters for CSOM for Class Incremental setting on CIFAR10

| Papers | Memory Size | Memory per Class | Model | epochs | Fixed Memory |
|---|---|---|---|---|---|
| Finetune, Replay, LwF, iCarl, EWC, Memo, Podnet, BiC | 2000 | 20 | Resnet32 | 1 | ✗ |

Table 13: Hyperparameters for benchmarks from PyCIL Zhou et al. (2023a) on CIFAR10

| Paper | Model | Data setting | lr | epochs | memory size |
|---|---|---|---|---|---|
| SCALE | Resnet18 | *seq* | 0.03 | 1 | 256 |

Table 14: Hyperparameters for benchmarks from SCALE Yu et al. (2022) using SGD optimizer

| | | | | | | | | | |
|---|---|---|---|---|---|---|---|---|---|
| 100.0 ± 0.0 | 0.0 ± 0.0 | 0.0 ± 0.0 | 0.0 ± 0.0 | 0.0 ± 0.0 | 0.0 ± 0.0 | 0.0 ± 0.0 | 0.0 ± 0.0 | 0.0 ± 0.0 | 0.0 ± 0.0 |
| 84.2 ± 1.55 | 76.88 ± 1.22 | 0.0 ± 0.0 | 0.0 ± 0.0 | 0.0 ± 0.0 | 0.0 ± 0.0 | 0.0 ± 0.0 | 0.0 ± 0.0 | 0.0 ± 0.0 | 0.0 ± 0.0 |
| 65.86 ± 1.37 | 73.09 ± 1.15 | 52.61 ± 2.5 | 0.0 ± 0.0 | 0.0 ± 0.0 | 0.0 ± 0.0 | 0.0 ± 0.0 | 0.0 ± 0.0 | 0.0 ± 0.0 | 0.0 ± 0.0 |
| 62.0 ± 1.42 | 67.05 ± 1.07 | 41.47 ± 2.12 | 46.31 ± 1.57 | 0.0 ± 0.0 | 0.0 ± 0.0 | 0.0 ± 0.0 | 0.0 ± 0.0 | 0.0 ± 0.0 | 0.0 ± 0.0 |
| 55.48 ± 1.41 | 65.26 ± 1.14 | 36.68 ± 1.92 | 43.8 ± 1.55 | 22.42 ± 1.16 | 0.0 ± 0.0 | 0.0 ± 0.0 | 0.0 ± 0.0 | 0.0 ± 0.0 | 0.0 ± 0.0 |
| 53.9 ± 1.59 | 63.2 ± 0.91 | 32.28 ± 2.28 | 32.22 ± 1.68 | 22.2 ± 0.42 | 31.49 ± 1.44 | 0.0 ± 0.0 | 0.0 ± 0.0 | 0.0 ± 0.0 | 0.0 ± 0.0 |
| 52.76 ± 1.34 | 61.82 ± 0.7 | 30.75 ± 2.13 | 29.7 ± 1.89 | 21.41 ± 0.83 | 31.01 ± 1.5 | 20.24 ± 1.21 | 0.0 ± 0.0 | 0.0 ± 0.0 | 0.0 ± 0.0 |
| 50.96 ± 0.77 | 59.95 ± 1.06 | 29.02 ± 1.74 | 28.08 ± 1.74 | 19.91 ± 0.96 | 29.31 ± 1.66 | 19.79 ± 1.17 | 37.3 ± 1.09 | 0.0 ± 0.0 | 0.0 ± 0.0 |
| 42.49 ± 1.07 | 53.13 ± 1.29 | 27.98 ± 1.74 | 27.06 ± 1.25 | 19.11 ± 1.05 | 28.64 ± 1.48 | 19.32 ± 1.05 | 37.14 ± 0.99 | 38.04 ± 1.61 | 0.0 ± 0.0 |
| 41.2 ± 1.11 | 42.96 ± 1.1 | 27.71 ± 1.59 | 26.19 ± 1.15 | 18.74 ± 1.12 | 28.51 ± 1.18 | 18.88 ± 0.96 | 36.25 ± 1.04 | 36.36 ± 1.65 | 36.01 ± 0.9 |

Table 15: mean and standard deviation of accuracies in a task matrix

## A.3 Domain Incremental Learning

For the domain incremental setting, we set number of tasks = 5, number of classes/task = 2

### A.3.1 MNIST

**Hyperparameters:** Table 16 contains the initial hyperparameters used for obtaining the results shown in Table 1b.

**Task matrices:** Table 17 indicates the resultant accuracy values obtained after performing 10 trials.

| Model | $H$ | $D$ | $\sigma_0$ | $\lambda_0$ | $\omega_h^2$ | $\lambda_\omega^0$ | $\tau_\sigma$ | $\tau_\lambda$ | p | s |
|---|---|---|---|---|---|---|---|---|---|---|
| Vanilla SOM | $15 \times 15$ | $28 \times 28$ | 0.6 | 0.07 | 1 | 0.9 | 8 | 45 | - | - |
| DendSOM | $8 \times 8$ | $7 \times [14 \times 14]$ | 4 | 0.95 | 2 | 0.005 | - | - | 10 | 3 |
| CSOM | $15 \times 15$ | $28 \times 28$ | 1.5 | 0.07 | 0.5 | 0.9 | 8 | 45 | - | - |

Table 16: Hyperparameters for Domain Incremental setting on MNIST

| | | | | |
|---|---|---|---|---|
| $99.85 \pm 0.04$ | $45.59 \pm 5.94$ | $45.63 \pm 3.38$ | $68.99 \pm 4.3$ | $39.82 \pm 5.51$ |
| $99.04 \pm 1.04$ | $93.57 \pm 2.13$ | $61.77 \pm 3.29$ | $73.77 \pm 10.54$ | $45.9 \pm 4.1$ |
| $97.55 \pm 1.65$ | $93.25 \pm 2.75$ | $93.88 \pm 2.96$ | $55.48 \pm 7.08$ | $18.41 \pm 3.85$ |
| $98.1 \pm 1.65$ | $92.6 \pm 2.78$ | $90.52 \pm 5.21$ | $96.94 \pm 0.97$ | $29.71 \pm 4.53$ |
| $97.99 \pm 1.65$ | $91.85 \pm 2.77$ | $77.5 \pm 9.88$ | $97.2 \pm 0.87$ | $80.92 \pm 6.1$ |

Table 17: mean and standard deviation of accuracies in a task matrix

### A.3.2 KMNIST

**Hyperparameters:** Table 18 contains the initial hyperparameters used for obtaining the results shown in Table 1b.

| Model | $H$ | $D$ | $\sigma_0$ | $\lambda_0$ | $\omega_h^2$ | $\lambda_\omega^0$ | $\tau_\sigma$ | $\tau_\lambda$ | p | s |
|---|---|---|---|---|---|---|---|---|---|---|
| Vanilla SOM | $20 \times 20$ | $28 \times 28$ | 0.6 | 0.07 | 1 | 0.9 | 8 | 45 | - | - |
| DendSOM | $12 \times 12$ | $7 \times [14 \times 14]$ | 6 | 0.95 | 2 | 0.005 | - | - | 4 | 2 |
| CSOM | $35 \times 35$ | $28 \times 28$ | 1.5 | 0.07 | 0.5 | 0.9 | 8 | 45 | - | - |

Table 18: Hyperparameters for Domain Incremental setting on KMNIST

**Task matrices:** Table 19 indicates the resultant accuracy values obtained after performing 10 trials

### A.3.3 Fashion-MNIST

**Hyperparameters:** Table 20 contains the initial hyperparameters used for obtaining the results shown in Table 1b.

**Task matrices:** Table 21 indicates the resultant accuracy values obtained after performing 10 trials.

### A.3.4 CIFAR-10

**Hyperparameters:** Table 22 shows hyperparameters used for training CSOM on CIFAR10 in a Domain Incremental Setting.

| | | | | |
|---|---|---|---|---|
| $98.6 \pm 0.4$ | $29.35 \pm 1.24$ | $54.72 \pm 2.57$ | $28.63 \pm 2.45$ | $47.21 \pm 2.47$ |
| $92.19 \pm 2.46$ | $91.48 \pm 1.68$ | $41.84 \pm 3.3$ | $55.72 \pm 3.25$ | $42.46 \pm 2.05$ |
| $89.91 \pm 2.75$ | $89.74 \pm 1.99$ | $89.85 \pm 2.09$ | $52.24 \pm 2.62$ | $39.38 \pm 1.62$ |
| $86.8 \pm 2.5$ | $89.55 \pm 1.94$ | $88.32 \pm 2.21$ | $93.01 \pm 1.83$ | $45.63 \pm 1.43$ |
| $86.1 \pm 2.69$ | $88.89 \pm 2.32$ | $86.72 \pm 3.45$ | $91.91 \pm 2.87$ | $86.92 \pm 2.3$ |

Table 19: mean and standard deviation of accuracies in a task matrix

| Model | $H$ | $D$ | $\sigma_0$ | $\lambda_0$ | $\omega_h^2$ | $\lambda_\omega^0$ | $\tau_\sigma$ | $\tau_\lambda$ | p | s |
|---|---|---|---|---|---|---|---|---|---|---|
| Vanilla SOM | $20 \times 20$ | $28 \times 28$ | 0.6 | 0.07 | 1 | 0.9 | 8 | 45 | - | - |
| DendSOM | $10 \times 10$ | $7 \times [14 \times 14]$ | 5 | 0.95 | 2 | 0.005 | - | - | 8 | 4 |
| CSOM | $25 \times 25$ | $28 \times 28$ | 1.5 | 0.07 | 0.5 | 0.9 | 8 | 45 | - | - |

Table 20: Hyperparameters for Domain Incremental setting on Fashion-MNIST

| | | | | |
|---|---|---|---|---|
| $96.72 \pm 1.44$ | $53.15 \pm 6.27$ | $28.88 \pm 4.84$ | $40.7 \pm 2.94$ | $49.77 \pm 0.27$ |
| $92.63 \pm 1.79$ | $95.08 \pm 0.61$ | $41.42 \pm 1.21$ | $44.46 \pm 1.51$ | $47.21 \pm 2.26$ |
| $92.94 \pm 1.72$ | $92.22 \pm 1.61$ | $97.55 \pm 0.87$ | $95.75 \pm 1.36$ | $83.07 \pm 3.94$ |
| $93.72 \pm 1.71$ | $91.37 \pm 1.49$ | $97.58 \pm 0.92$ | $97.22 \pm 0.86$ | $86.17 \pm 4.71$ |
| $93.96 \pm 1.73$ | $90.77 \pm 2.15$ | $97.1 \pm 0.85$ | $97.06 \pm 1.22$ | $99.13 \pm 0.13$ |

Table 21: mean and standard deviation of accuracies in a task matrix

| Model | $H$ | $D$ | $\sigma_0$ | $\lambda_0$ | $\omega_h^2$ | $\lambda_\omega^0$ | $\tau_\sigma$ | $\tau_\lambda$ |
|---|---|---|---|---|---|---|---|---|
| CSOM | $15 \times 15$ | $28 \times 28$ | $1.5$ | $0.07$ | $0.5$ | $0.9$ | $8$ | $45$ |

Table 22: Hyperparameters for Domain Incremental setting on CIFAR10

Table 23 shows task matrix of accuracy values obtained after performing 8 trials of CSOM on grayscale images from CIFAR-10.

| | | | | |
|---|---|---|---|---|
| $99.52 \pm 0.37$ | $46.01 \pm 4.66$ | $43.63 \pm 5.55$ | $65.38 \pm 5.11$ | $38.67 \pm 6.55$ |
| $98.79 \pm 0.59$ | $93.15 \pm 1.09$ | $60.14 \pm 2.78$ | $74.98 \pm 7.41$ | $50.55 \pm 2.84$ |
| $97.15 \pm 1.38$ | $91.67 \pm 1.84$ | $94.41 \pm 1.08$ | $52.1 \pm 7.59$ | $18.99 \pm 3.8$ |
| $95.53 \pm 7.62$ | $91.35 \pm 1.74$ | $89.13 \pm 5.01$ | $95.36 \pm 0.72$ | $30.34 \pm 3.01$ |
| $97.25 \pm 2.91$ | $91.23 \pm 0.92$ | $68.91 \pm 5.9$ | $95.85 \pm 0.77$ | $79.57 \pm 5.46$ |

Table 23: mean and standard deviation of accuracies in a task matrix

