# OpenReview forum: "Neuro-mimetic Task-free Unsupervised Online Learning with Continual Self-Organizing Maps"
_TMLR — Rejected by TMLR_

### Review · Reviewer_tAPL · 2025-05-16

**Summary Of Contributions:**

This paper studies continual unsupervised learning, an area that has received much less attention than continual supervised learning. More specifically, the authors focus on a particular type of unsupervised learning algorithm, the self-organising map (SOM), and study the online multitask setting, where samples from a set of discrete tasks are seen sequentially and only once.

The main contributions of the paper are:

* evaluation of the native continual learning abilities of SOMs when adapted to the online multitask setting (on a set of standard supervised tasks for continual learning)
* extension of SOMs to better handle the stability-plasticity tradeoff in a continual setting by incorporating a second weight matrix that measures the running variance of each unit (and empirical proofs that this does indeed help)

**Audience:**

Yes

**Broader Impact Concerns:**

No.

**Claims And Evidence:**

Yes

**Requested Changes:**

* [critical] List all the modifications of CSOM w.r.t. SOM and based on it, clearly explain the role of each extra component in CSOM and its contribution to CL:
    * 1) in the text, by make it clearer (even if some short explanations are already present) what the motivation and the role behind each component is
    * 2) in the experiments, by running at least some ablation studies (e.g. why is the running variance necessary? what happens if the binary mask is not present? what happens if the radius and learning rate are global parameters?, why is the $\delta$ in line 4 of Algorithm 2 needed?) and seeing the effects both in terms of performance and qualitatively in terms of unit activations (i.e. supporting the claims in 5.5 regarding recruitment of untrained units for new tasks). Also showing the dependence of certain components w.r.t others (e.g. visualisation of relationship of neighbourhood function with running variance, and explicit explanation of how this helps with CL) could help better understand.


* [would improve paper] For clarity and readability, consider writing in Algorithm 2 a simpler distilled version of CSOM, hiding away some details in the Appendix.
* [would improve paper] Can you provide an estimate of time/space complexity increases incurred in CSOM compared to SOM, and potentially compare to some supervised baseline like EWC for reference?
* Minor:
    * Table 1 legend: add explanation of what the three column/tasks are (potentially move this hyper parameter list to appendix, as it is not very informative in main text, or else explain why is it important to have it there?)
    * Backward transfer quantification: in Eq. (8), BWT seems to be measuring differences between accuracy in task $T$ and in task $i$ after learning task $i$. If BWT measures the amount of improvement that a new task T has on the previously learned tasks (as the text states and as is commonly accepted), shouldn’t it be computed based on $a_{i, T} - a_{i, i}$ similar in spirit to the forgetting measure? It’s not fully clear to me what this measure captures and why lower is better? Note also that Table 2 highlights higher (not lower) values. Maybe I am misreading something.

**Strengths And Weaknesses:**

Strengths:

* Flexible proposed method: the proposed continual-SOM (CSOM) is relatively simple and compared to other solutions does not require maintaining samples from previously observed tasks.
* Convincing results: CSOM significantly outperforms the standard SOM in a basic set of continual learning tasks and performs competitively with a range of continual supervised learning algorithms in the more challenging Split-CIFAR dataset.
* Interesting topic: explores continual learning in the unsupervised setting, which is an important topic that has been much less studied than its supervised counterpart.
* Clarity: the paper is nicely written, relatively clear and concise. In general it is easy to follow (see caveats below).
* Theoretical analysis: the authors provide a mathematical analysis on the update rule implemented in CSOM and provide theoretical arguments on the origin of CSOM’s continual learning abilities

Weaknesses:

* CSOM components are not well motivated: compared to SOM, CSOM includes a set of extra components and steps (such as the variance matrix, the binary mask for updates, the neuron-specific radius and learning rate, the more complex neighbourhood function...) and it is not fully clear from the text nor the experiments what each of the extra components brings relative to the others. Also, even though the basic idea behind CSOM is not necessarily complex, the core of Algorithm 2 gets obscured by many minor details that make the notation quite cumbersome.
* Evaluation is performed only on supervised tasks: even though the explored algorithm is unsupervised. This is valuable to assess how CSOM compares to other SOTA in these tasks, but other (potentially simpler) algorithms exist for this type of problems. It would have been nice to see some results in use cases where the unsupervised nature of SOMs is key, or where its basic generative abilities can be used.
* Certain statements are not directly supported by experimental or theoretical evidence (e.g. in 5.5, it is never shown that indeed adding the running variance helps recruit untrained units when facing new tasks).
* Although theoretical proofs of the algorithm’s convergence and stability are welcome, the proofs and derivations and the conclusions made from them seem a little hand-wavy at some points.

---

> ### Author Response · Authors · 2025-08-04
> **Part 1.1 - response to comment 1**
>
> Thank you for the opportunity to clarify our approach. ,
>
> Notice that, first of all, the CSOM maintains an additional (non-negative) matrix  $\\mathbf{M}^{\\omega^2} \\in \\mathcal{R}^{D \\times H}, (\mathcal{R} > 0)$, which contains synaptic weight parameters associated with the ``running variance'' ($\omega^2$) of each neuronal unit in the system. This means that each unit/prototype in the CSOM is defined by two vectors of weights -- one for approximate unit means and another for approximate standard deviations.
> The intuition is that each unit in our SOM model is generalized to maintain its own learnable multivariate Gaussian distribution (inspired by the latent variables of incremental Gaussian mixture models) with a diagonal covariance matrix. We denote the initial scaling/update factor for running variance of every unit as $\lambda^0_{\omega}$ as shown in step 11 of Algorithm 3 (Appendix). The running variance is required in the distance function between CSOM units and the input sample, to identify empty units whenever the distribution of input samples change. It thus prevents semantic overlap of newer inputs over units trained on previous tasks. In SOMs, the ratio of weight update that each connection goes through depends on its cartesian distance from the BMU in a topology. In CSOM, we empirically found that the update/scaling factor of running variance of every neuron is proportional to the magnitude of its weight update. Therefore, this update factor for a neuron is decayed as per its distance from the BMU to obtain $\lambda_{\omega}$ in step 12 of Algorithm 3. The running variance vector coupled to every neuron in the CSOM can be used to generate samples belonging to the class that matched it. Thus, the underlying structure of our neural system could be likened to a simple, dynamic generative model.
>
> In addition to the local running variance parameters, the CSOM is designed to promote a form of neuronal competition driven by unit-centric learning and distance weighting parameters. \textbf{Specifically, each neuron $h \in H$ arranged in topology $\mathcal{G}$ is assigned an independently-controlled, dynamic radius $\sigma_h$ and learning rate $\lambda_h$ parameter (as shown in Figure 3}. As shown in Algorithm 2, in the routine \textbf{Update()}, we particularly decay learning parameters only for the BMU ($u$), i.e., only $\sigma_u$ and $\lambda_u$ are decayed at time $t$. This localized decay is furthermore a function of the number of times that unit $u$ has been selected as the BMU, i.e., unit $u$ adjusts $\sigma_u$ and $\lambda_u$ as a function of its BMU count $\eta_u$. We do not decay $\sigma_u$ and $\lambda_u$ after they reach certain infinitesimally small threshold to ensure their values do not plummet to 0 and some positive weight update is always achieved as $\eta_u \rightarrow T$. The neighborhood function of the CSOM is also notably a function of the running variance parameters $\mathbf{M}^{\omega^2}$ and $\delta$ (step 4 - Algorithm 3), further facilitating the calculation of a per-unit region of influence (treating each neuron as its own weighted multivariate Gaussian distribution). As a result, whenever a weight update is triggered, synaptic values are adjusted on a linear scale for the BMU $u$ while, for any non-BMU units $v_j$, the update is adjusted on a scale that decreases as the euclidean distance from $u$ increases. The $\delta$ parameter gives a gaussian nature as shown in Figure 1 to the neighborhood function which is needed for fading weight updates in the surrounding of BMU. Without $\delta$, the formulation of even vanilla SOM fails.

---

> > ### Author Response · Authors · 2025-08-04
> > **Part 1.2 - response to comment 1**
> >
> > Experimentally, we noticed that the Gaussian function used to scale the synaptic weight updates in a neighborhood was liable to cause leaky weight updates in the units that were very far away from the BMU in topology. This caused an explosion of weight update in the available untrained units, thus undesirably increasing the size of cluster for current incoming task. To control this behavior, we enforced a hard bound over the synaptic weight update in the neighborhood using the current $\sigma_u$ value of the BMU. For this we create a binary mask as shown in step 6 of **Update()** subroutine in Algorithm 3. Since the goal of CSOM is to have linearly separable trained clusters of streaming input data, it is required that the radius and learning rate also remain independent for different classes/tasks. A global copy of radius and learning rate set to high value initially leads to larger clusters for initial classes/tasks passed in the streaming setting and smaller clusters for the ones appearing later. This creates imbalanced clustering in the SOM as supported by our empirical trials. With each unit having its own copy of radius and learning rate in CSOM, every class/task gets the opportunity to form a cluster according to variance of its respective input samples. The running standard deviation used in the distance metric shown in line 18 of Algorithm 3 is decayed according to the number of times a unit has been selected as BMU. It reduces the chances of an already well trained unit to get selected again as BMU for similar looking sample but belonging to different class (eg, 3 and 8). An untrained or less trained neighboring unit will have relatively higher running standard deviation thus giving a lower distance value from the input sample. This promotes formation of cluster in the empty space of CSOM creating better quality of trained clusters towards the end as seen in Figure 1 and 2. Also, this follows the principle of competitive learning among the CSOM units leading to better performance and faster convergence than vanilla SOM. This distance function also makes CSOM immune to sudden changes in the distribution of input without task boundary heuristic. Further details on hyperparameter tuning is provided in the appendix.
> >
> > The input is tiled to match the dimensions of number of units in topology $\mathcal{G}$ of CSOM, as a result, it can perform distance and BMU calculation in parallel for all CSOM units. It give CSOM a 3x speedup over vanilla SOM.
> >
> > Figures 6, 7, and 8 show relation between parameters and their consequent metrics obtained using Fashion MNIST dataset. We empirically found that increasing learning rates leads to faster adaptation of CSOM to the innput data but may have chance of instability or overfitting. There may be semantic overlap due to high learning rate values for CSOM units or the resultant cluster of a class may not capture good variance of input samples in CSOM. Low learning rate values lead to slower convergence. CSOM may be more stable in this case but may also lead to underfitting if the learning rate value is too low. Higher radius values of CSOM units leads to broader neighborhood influence that is bigger clusters of classes from input data distribution. Lower radius values on the other hand lead to lesser interference on the line of separability between clusters.
> > A grid search with CSOM over all datasets indicated that initial running variance has to be just a little higher than the variance of the input distribution to get distinct clusters in the trained model. At the same time, it is important to keep the initial radius value just $>1$ but not higher than $2$ because that gives more spread for weight update to every BMU thus failing to capture a good compressed distribution of input samples in the finally trained CSOM. Since the distribution of $\mathbf{M}^{\omega^2}$ is just higher than input distribution and its updates are tied with the weight update of $\mathbf{M}$, the initial $\lambda_{\omega}^0$ also has to be higher than $\lambda_0$. This can be seen from their choices mentioned in Tables 5, 6, 8, 10, 12, 16, 18, 20, and 22.
> > The scaling factors $\tau_{\sigma}$ and $\tau_\lambda$ contribute less to learning compared to $\sigma$ and $\lambda$ as evident from the correlation matrix in Figure 8. Experimentally we have found that values in the range of 7-9 work well for $\tau_\sigma$ and 40-45 for $\tau_\lambda$.

---

> > > ### Author Response · Authors · 2025-08-04
> > > **Part 1.3 - response to comments 2, 3, 4, 5**
> > >
> > > ## 2. Rewrite Algorithm 2
> > > The algorithm 2 is rewritten with more clarity on page 5 and the pseudocode is move to the appendix
> > >
> > > ## 3. Time and Space complexity of CSOM
> > > The L2 distance along with normalization by the running standard deviation for every input sample has time complexity of O($D \times H$). Computing the arg min has O($H$) time complexity whereas, weight update is again O($D\times H$). Therefore, for $N$ input samples, the overall time complexity of CSOM will be O($N\times D\times H$). At runtime, we maintain the CSOM weights and their respective running variance values (both, O($D\times H$)) along with the input sample of size ($D$), therefore the total space complexity remains O$(D + 2(D\times H))$ which is, O($D\times H$)
> > >
> > > ## 4. Table 1 explanation and placement in the paper
> > > Table 1 in the original draft is now moved to appendix as Table 5. The parameters can be described as follows,
> > > $\sigma_0 \rightarrow$ initial radius for every unit \
> > > $\lambda_0 \rightarrow$ initial learning rate for every unit \
> > > $\omega_0^2 \rightarrow$ initial running variance for every unit \
> > > $\lambda_0^2 \rightarrow$ initial update factor for running variance
> > >
> > > ## 5. Explanation for Backward transfer
> > > BWT is calculated based on the accuracy matrix which few papers consider in row major form and other in column major form. We provide citations to papers like GEM in Section 5.4 which talk about BWT. We have corrected the interpretation of BWT stating "higher is better". Therefore, if the BWT score is negative, then a lower magnitude of BWT value is preferred. A model with high BWT value will experience improvement in identification of a task in the past as an effect of good learning on the current task.

---

### Review · Reviewer_aD8n · 2025-05-26

**Summary Of Contributions:**

This work addresses the critical challenge of catastrophic forgetting, specifically focusing on self-organizing maps (SOMs). While catastrophic interference has been extensively studied in deep neural networks for supervised learning, there has been limited investigation into unsupervised neural systems like SOMs operating in lifelong learning contexts.

The authors adapted the classical SOM model from its original single static dataset formulation to online continual learning settings, conducting systematic experiments to measure memory retention capabilities alongside adaptation to sequential pattern streams. The work demonstrated that despite theoretical expectations, SOMs experience substantial catastrophic interference across all examined benchmarks. This involved introducing specialized decay functionality to mitigate forgetting and implementing running variance mechanisms to improve best matching unit (BMU) selection over time. The CSOM represents the first systematic attempt to create a SOM variant that maintains the model's core unsupervised competitive learning principles while addressing catastrophic forgetting through principled architectural modifications. The paper conducted comprehensive experimental validation across four class-incremental datasets (MNIST, Fashion-MNIST, Kuzushiji-MNIST, and CIFAR-10). They performed systematic comparisons between standard SOMs and the proposed CSOM, demonstrating significantly reduced forgetting in the CSOM across all benchmarks. These experiments provide robust evidence that the CSOM successfully addresses catastrophic forgetting while maintaining the desirable properties of SOMs for clustering and dimensionality reduction tasks.

**Audience:**

Yes

**Broader Impact Concerns:**

I have no concerns regarding the ethical implications of the work

**Claims And Evidence:**

Yes

**Requested Changes:**

The following are some of the suggestions that would enhance the paper's robustness and readability.

The authors' claim that "Although forgetting in the context of ANNs has been studied extensively, there still exists far less work investigating it in terms of unsupervised architectures such as the venerable self-organizing map (SOM), a neural model often used in clustering and dimensionality reduction" does not constitute a compelling justification for using SOMs in continual learning. This assertion becomes particularly questionable given the poor experimental results obtained. The authors need to provide stronger theoretical or empirical motivation for why SOMs would be advantageous in this context.

Both Figure 1 and Figure 2 suffer from poor visual quality and unclear content presentation. These figures require complete redesign to effectively communicate their intended message and meet publication standards.

The paper contains inconsistent notation usage. The time step variable t is introduced in the text stating "time steps are tracked in simulation within the variable t" but does not appear in the preceding mathematical formulation. Furthermore, the same symbol t is used in Algorithm 1 with a different meaning, creating confusion for readers.

The methodological section is inadequately developed, providing insufficient explanation of the proposed approach. Readers must rely on the algorithm to understand the method, but the algorithm itself is difficult to interpret and lacks clarity in its presentation.

The mathematical analysis section should be substantially condensed, with detailed proofs relocated to appendices. The current extensive mathematical treatment impedes the paper's flow and obscures more important content.

The experimental section suffers from misplaced emphasis, dedicating excessive space to setup details such as dataset composition, parameter specifications, and evaluation protocols while providing insufficient discussion of the actual results and their implications.
The paper lacks essential ablation studies and parameter sensitivity analyses. The method's parameters are merely listed in Table 1 without proper investigation of their impact on performance or guidance for their selection.

Given these substantial issues, I believe the paper requires comprehensive rewriting to address these fundamental concerns and improve its overall quality.

**Strengths And Weaknesses:**

The paper addresses an understudied problem in continual learning: unsupervised continual learning. The authors propose an interesting approach using a modified version of self-organizing maps (SOMs) that reduces catastrophic forgetting while adapting to streaming data.
However, the paper suffers from several significant limitations. First, the motivation for using SOMs is unclear, as the authors fail to provide comparisons with other established continual learning approaches. This omission makes it difficult to assess the relative benefits of the proposed method.

Second, the paper's presentation is problematic, with emphasis placed on less critical aspects of the work. The mathematical analysis receives disproportionate attention compared to the core methodology, which is only briefly explained. Readers must rely primarily on Algorithm 2 to understand the approach, but this algorithm is difficult to follow and contains unclear elements. For instance, the meaning of λ_v in line 7 is not defined, and the tilde notation is confusing throughout.

Third, the experimental evaluation is inadequate. Not only does the method achieve poor performance across all benchmarks, but it also lacks comparison with relevant baseline methods. While Table 3 presents comparisons between CSOM and some continual learning approaches, none of these baseline methods were specifically designed for online continual learning contexts. A fair evaluation would require comparisons with methods developed for similar online settings, even if they employ fundamentally different architectures.

Finally, the overall quality of writing requires substantial improvement throughout the manuscript. The paper would benefit from careful revision to enhance clarity, coherence, and technical precision.

---

> ### Author Response · Authors · 2025-08-04
>
> Thank you for raising questions and giving us an opportunity to clarify. Following is our response,
>
> ## 1. stronger theoretical or empirical motivation for why SOMs would be advantageous in this context
> The benchmark results along with the training scheme of CSOM indicates its merit in being a simple yet effective model that does not require any task descriptor and still maintains linear separability amongst task specific samples by virtue of its distance calculation. This makes it truly online and unsupervised in nature. The simple nature of this model makes it a good choice for memory augmentation of networks subject to further testing. CSOM still gives the best average accuracy among all other models even with unsupervised setting. This is also corroborated by our redesigned proofs.
>
> ## 2. Both Figure 1 and Figure 2 suffer from poor visual quality and unclear content presentation
> Figure 1 and 2 are now redesigned to give a better understanding of the concepts to readers on pages 5, 6, 7.
>
> ## 3. Inconsistent notation especially, $\textit{t}$ in algorithm 1
> The algorithm is now corrected on page 4 and the $\textit{t}$ has been removed to make things consistent. A specific equations are mentioned below for brevity, \
> $\lambda_t = \lambda_0 \exp{(-t / \tau_\lambda)}$\
> $\sigma_t = \sigma_0 \exp{(-t / \tau_\sigma)}$
>
> ## 4. The methodological Section is inadequately developed
> We have redesigned Algorithm 2 for better understanding of continual SOM (all the notations are summarized in Table 4, Figure 2), built to process and generalize dynamically from samples drawn from a stream of tasks. Notice that, first of all, the CSOM now maintains an additional (non-negative) matrix $\mathbf{M}^{\omega^2} \in \mathcal{R}^{D \times H}_{+}$, which contains synaptic weight parameters associated with the ``running variance'' ($\omega^2$) of each neuronal unit in the system. This means that each unit/prototype in the CSOM is defined by two vectors of weights -- one for approximate unit means and another for approximate standard deviations.

---

> > ### Author Response · Authors · 2025-08-04
> > **Part 2.2 of the response**
> >
> > ## 5. The methodological Section is inadequately developed
> > We have redesigned Algorithm 2 for better understanding of continual SOM (all the notations are summarized in Table 4, Figure 2), built to process and generalize dynamically from samples drawn from a stream of tasks. Notice that, first of all, the CSOM now maintains an additional (non-negative) matrix $\mathbf{M}^{\omega^2} \in \mathcal{R}^{D \times H}, (\mathcal{R} > 0)$, which contains synaptic weight parameters associated with the ``running variance'' ($\omega^2$) of each neuronal unit in the system. This means that each unit/prototype in the CSOM is defined by two vectors of weights -- one for approximate unit means and another for approximate standard deviations.
> > The intuition is that each unit in our SOM model is generalized to maintain its own learnable multivariate Gaussian distribution (inspired by the latent variables of incremental Gaussian mixture models) with a diagonal covariance matrix. Notice that, in the $\textbf{Update()}$ routine of our CSOM, the variance parameters are adjusted using a Hebbian-like rule inspired by Welford's online algorithm but modified to use the weighting provided by our CSOM's neighborhood function. We denote the initial scaling/update factor for running variance of every unit as $\lambda^0_{\omega}$ as shown in step 11 of Algorithm 2. In SOMs, the ratio of weight update that each connection goes through depends on its cartesian distance from the BMU in a topology. In CSOM, we empirically found that the update/scaling factor of running variance of every neuron is proportional to the magnitude of its weight update. Therefore, this update factor for a neuron is decayed as per its distance from the BMU to obtain $\lambda_{\omega}$ in step 11 of Algorithm 2. The running variance vector coupled to every neuron in the CSOM can be used to generate samples belonging to the class that matched it. Thus, the underlying structure of our neural system could be likened to a simple, dynamic generative model.
> >
> > In addition to the local running variance parameters, the CSOM is designed to promote a form of neuronal competition driven by unit-centric learning and distance weighting parameters. \textbf{Specifically, each neuron $h \in H$ arranged in topology $\mathcal{G}$ is assigned an independently-controlled, dynamic radius $\sigma_h$ and learning rate $\lambda_h$ parameter (as shown in Figure 3)}. As shown in Algorithm 2, in the routine $\textbf{Update()}$, we particularly decay learning parameters only for the BMU ($u$), i.e., only $\sigma_u$ and $\lambda_u$ are decayed at time $t$. This localized decay is furthermore a function of the number of times that unit $u$ has been selected as the BMU, i.e., unit $u$ adjusts $\sigma_u$ and $\lambda_u$ as a function of its BMU count $\eta_u$. We do not decay $\sigma_u$ and $\lambda_u$ after they reach certain infinitesimally small threshold to ensure their values do not plummet to 0 and some positive weight update is always achieved as $\eta_u \rightarrow T$. The neighborhood function of the CSOM is also notably a function of the running variance parameters $\mathbf{M}^{\omega^2}$ and $\delta$ (step 4 - Algorithm 3), further facilitating the calculation of a per-unit region of influence (treating each neuron as its own weighted multivariate Gaussian distribution). As a result, whenever a weight update is triggered, synaptic values are adjusted on a linear scale for the BMU $u$ while, for any non-BMU units $v_j$, the update is adjusted on a scale that decreases as the euclidean distance from $u$ increases. The CSOM utilizes a masking matrix ($\mathbf{o}$) to control the scale of its synaptic weight updates. The mask matrix also prevents any leaky weight updates caused by infinitesimally small values passed by the neighborhood function ($\phi$).
> >
> > Crucially, the proposed CSOM is a task-free model, which means that it does not require any information about task boundaries (in the form of task descriptors).

---

> > > ### Author Response · Authors · 2025-08-04
> > > **Part 2.3.a of the response**
> > >
> > > ### 7. The experimental section suffers from misplaced emphasis, dedicating excessive space to setup details such as dataset composition, parameter specifications, and evaluation protocols while providing insufficient discussion of the actual results and their implications. The paper lacks essential ablation studies and parameter sensitivity analyses. The method's parameters are merely listed in Table 1 without proper investigation of their impact on performance or guidance for their selection
> > >
> > > **Response**
> > > Experimentally, we noticed that the Gaussian function used to scale the synaptic weight updates in a neighborhood was liable to cause leaky weight updates in the units that were very far away from the BMU in topology. This caused an explosion of weight update in the available untrained units, thus undesirably increasing the size of cluster for current incoming task. To control this behavior, we enforced a hard bound over the synaptic weight update in the neighborhood using the current $\sigma_u$ value of the BMU. For this we create a binary mask as shown in step 6 of \textbf{Update()} subroutine in Algorithm 3 (Appendix). Since the goal of CSOM is to have linearly separable trained clusters of streaming input data, it is required that the radius and learning rate also remain independent for different classes/tasks. A global copy of radius and learning rate set to high value initially leads to larger clusters for initial classes/tasks passed in the streaming setting and smaller clusters for the ones appearing later. This creates imbalanced clustering in the SOM as supported by our empirical trials. With each unit having its own copy of radius and learning rate in CSOM, every class/task gets the opportunity to form a cluster according to variance of its respective input samples. The running standard deviation used in the distance metric shown in line 12 of Algorithm 2 is decayed according to the number of times a unit has been selected as BMU. It reduces the chances of an already well trained unit to get selected again as BMU for similar looking sample but belonging to different class (eg, $3$ and $8$). An untrained or less trained neighboring unit will have relatively higher running standard deviation thus giving a lower distance value from the input sample. This promotes formation of cluster in the empty space of CSOM creating better quality of trained clusters towards the end as seen in Figure 4 and 5. Also, this follows the principle of competitive learning among the CSOM units leading to better performance and faster convergence than vanilla SOM. This distance function also makes CSOM immune to sudden changes in the distribution of input without task boundary heuristic. Figures 6, 7, 8 in appendix show analysis on parameter sensitivity of CSOM.
> > >
> > >     In Table 1, we show benchmark comparison of CSOM with other variants on grayscale images datasets. DendSOM, is a variant of SOM designed for continual learning setting but fails to perform well in online unsupervised setting as seen from its result values. In Table 2, we present a comparison of model performance measurements on the split-CIFAR10 benchmark. Among all the benchmarks, SCALE particularly had the exact same setting (shown in Table 2) as ours and had state-of-the-art (SOTA) ACC values in a semi-supervised learning setting. Although it had marginally higher mean LA value, their standard deviation was almost $7$ times higher than our CSOM after $10$ trials. Due to the online setting of single epoch and very low buffer storage, few algorithms experience more catastrophic forgetting thus giving us close to $0$ standard deviation in their evaluation over ($n=10$) trials. Except for EWC, all of the models used convolution and most of them used a storage buffer (memory). Nevertheless, our CSOM model yielded the best average accuracy among all others without maintaining any buffer storage or requiring convolution to obtain better performance, despite operating in an online setting (epochs=$1$, each input sample processed once).
> > >
> > >     As shown in Figure 4, as a result of the neural clusters formed, our model appeared to acquire a good latent representation of every input task. This means that there is a good amount of variance among the trained units allocated for every task. However, despite our good experimental results, we could not obtain an equal number of trained units per class. Moreover, some neuronal units (especially those at the line of separability of clusters) have fuzzy representations. This may occur if a neuronal unit receives a weight update that corresponds to multiple different classes.

---

> > > > ### Author Response · Authors · 2025-08-04
> > > > **Part 2.3.b of response**
> > > >
> > > > ## 7. The experimental section suffers from misplaced emphasis, dedicating excessive space to setup details such as dataset composition, parameter specifications, and evaluation protocols while providing insufficient discussion of the actual results and their implications. The paper lacks essential ablation studies and parameter sensitivity analyses. The method's parameters are merely listed in Table 1 without proper investigation of their impact on performance or guidance for their selection
> > > >
> > > > Nevertheless, as indicated in Table 1 and 2, **the CSOM achieves the best performance compared to all of the other models/variants of SOM on all four benchmarks/datasets and successfully beats the SOTA model on split-CIFAR10 dataset**. This empirical result indicates that our CSOM framework, leveraging a competitive learning scheme, can yield the potentially best memory retention (or greatest reduction in forgetting) when processing classes incrementally from a data stream. The visualization of the samples synthesized for this particular model -- see Figure 4 -- also corroborate this result qualitatively; the samples, in this case, look the clearest and the model's internal units seem to represent most of the individual classes/tasks (including those presented at the start of the task sequence).

---

> > > > > ### Author Response · Authors · 2025-08-04
> > > > > **Part 2.4 of the response**
> > > > >
> > > > > ## 6. The mathematical analysis section should be substantially condensed
> > > > > **Response**
> > > > > We have rewritten our proofs with the reviews in mind. Requesting you to please check page 7-10. We could not write them here due to latex compatibility issues.

---

### Review · Reviewer_MW3m · 2025-07-13

**Summary Of Contributions:**

In this paper, the author introduces CSOM, a variant of the classical SOM augmented with per-neuron learning–rate/radius schedules and a running-variance matrix. CSOM is designed for task-free, one-pass, unsupervised class-incremental learning and is claimed to reduce catastrophic forgetting on four vision benchmarks (MNIST, Fashion-MNIST, KMNIST, split-CIFAR-10) compared with both vanilla SOMs and popular supervised continual-learning baselines.

**Audience:**

Yes

**Broader Impact Concerns:**

No.

**Claims And Evidence:**

No

**Requested Changes:**

Please refer to my comments above.

**Strengths And Weaknesses:**

**Strengths:**

S1. Catastrophic forgetting in unsupervised models is far less studied than in supervised deep nets; adapting SOMs to online class-incremental streams fills a genuine gap

S2. The per-unit running variance and neighbourhood-scaled Hebbian updates extend SOMs without introducing back-propagation or replay buffers, keeping the algorithm lightweight and biologically inspired.


**Weaknesses:**

W1. Algorithm 2 lines 14–15 update the BMU’s parameters as $σ_u^{(t)}=σ_u^{(t-1)}·exp( η_u / τ_σ )$ and $λ_u^{(t)} = λ_u^{(t-1)}·exp( η_u / τ_λ )$. With *positive* $τ$, these formulas increase $σ$ and $λ$ every time the unit wins, the opposite of the intended “decay”. Exponential growth of σ widens the neighborhood indefinitely and growth of $λ$ pushes weight updates toward instability. Subsequent convergence and stability claims become void.

W2. $λ_ω$ is recomputed per neuron as $(λ^{0}_ω − 0.5) + sigmoid(·)$ (Alg. 2 line 11). Depending on $λ^{0}_ω$ this expression can exceed 1 or become < 0, violating the assumption 0 < $λ_ω$ < 1 used in Lemma 4.1.

W3. Some “proof”s in this paper are qualitative, but not formal mathematical proof. For instance, in the proof of Theorem 4.3 (equilibrium), it restates design intentions (“careful balance”, “typically decreases”) but provides no formal bounds linking $λ_ω, λ_i, σ_u$ and $ϕ$ to convergence. Given the flaws above, equilibrium is presently unsubstantiated.

W4. There are internal inconsistencies in this paper. For example, $λ_i$ is described as “fixed learning rate” in Proposition 4.2 but defined as adaptive in Algorithm 2.

W5. Many cells show 0 or 1.7 × $10^{-15}$ after n = 10 trials (e.g. Replay LA, iCaRL ACC). Such tiny variance is unrealistic even with identical seeds. Is there any potential explanation on this phenomena?

W6. The study claims state-of-the-art online continual-learning performance yet compares CSOM mainly to offline, supervised CL methods (iCaRL, EWC, PODNet, etc.); I found only one true OCL baseline, SCALE, while standard OCL baselines like DER++, SSOCL (Imai et al., 2024) and R2R (Replay-to-Remember, 2024) are omitted.

W7. Variance-aware distance metrics and per-unit adaptive radii have appeared in earlier extensions of SOMs and Gaussian-mixture SOM variants; CSOM mainly combines existing heuristics rather than introducing a fundamentally new learning principle.

---

> ### Author Response · Authors · 2025-08-04
> **Part 1 of the response**
>
> Thank you for reviewing our work and raising important concerns. We will try to address them in our response below.
>
> ### 1. Growth or decay or $\sigma_u$ and $\lambda_u$
> **Response:**
> Thank you for pointing out the correction. The decay functions did not have a negative symbol. They should be $\sigma_u = \sigma_u * \exp{(-\eta_u / \tau_{\sigma})}$ and $\lambda_u = \lambda_u * \exp{(-\eta_u / \tau_{\lambda})}$. This has been updated in Algorithm 2 as well as pseudocode given in the Appendix.
>
> ### 2. $\lambda_{\omega}$ is recomputed per neuron as $(\lambda^0_{\omega} - 0.5) + sigmoid(.)$ (Alg. 2 line 11). Depending on $\lambda_{\omega}^0$ this expression can exceed 1 or become < 0, violating the assumption $0 < \lambda_{\omega} < 1$ used in Lemma 4.1. $\lambda_{\omega}$ can exceed 1
> **Response:**
> As shown in pseudocode, $sigmoid(.)$ gets the input of $-unit\_distance/\tau_1$. $\tau_1$ is calculated in runtime at line 5 in Algorithm 3 in the Appendix. If we assume the initial parameter setting mentioned in Table 5 and substitute them in the equation for $\lambda_{\omega}$ then we get, \\
>  $$\lambda_{\omega} = \left(0.9-0.5\right)\ +\ \frac{1}{\left(1\ +\ \exp\left(-\frac{x}{2\cdot(1.2)^{2}\cdot\log\left(\frac{10^{-8}}{0.07}\right)}\right)\right)} $$
>
> We plot this equation on a graph as shown in Figure 9 on page 26 of the main document to prove that the overall value never exceeds 1.
>
> ### W5. Very small values of standard deviation in benchmarks
> **Response**
> We regenerated the benchmarks of concern using libraries like PyCil and Mammoth to obtain better values of standard deviation. You can find them in Table 2.
>
> ### W6. The study claims state-of-the-art online continual-learning performance yet compares CSOM mainly to offline, supervised CL methods (iCaRL, EWC, PODNet, etc.); I found only one true OCL baseline, SCALE, while standard OCL baselines like DER++, SSOCL (Imai et al., 2024) and R2R (Replay-to-Remember, 2024) are omitted.
> **Response**
> DendSOM, SCALE and DER++ use similar setting as CSOM and we have provided benchmark results on all of them to provide a fair comparison. We contacted the authors of SSOCL and R2R but their code is not public yet at the time of this rebuttal submission.

---

> > ### Author Response · Authors · 2025-08-04
> > **Part 2 of the response**
> >
> > ## W3 and W4 concerns about proofs
> > **Response**
> > We have rewritten our proofs with the reviews in mind. Requesting you to please check page 7-10. We could not write them here due to latex compatibility issues.

---

### Decision · Action_Editor_Cge5 · 2025-09-10

**Recommendation:** Reject

**Audience:**

No

**Audience Explanation:**

The general problem setting is important and interesting, but the major issues identified in the reviews would make this paper less interesting to TMLR's audience.

**Claims And Evidence:**

No

**Claims Explanation:**

This paper addresses an important and under-explored problem, that of task-free unsupervised continual learning. The reviewers were generally positive on the problem's relevance, and the algorithm’s lightweight and biologically-inspired nature. However, they identified major concerns with errors in the approach and proofs (including the rewritten versions) that undermine the technical claims, issues in the experimental analysis and results, and general issues of clarity, organization, and correctness that indicate that this article is not yet ready for publication. These issues are unresolved in the revision provided in response to the initial reviews, and the reviewers universally recommend rejection.